# Speculative Decoding with Big Little Decoder

**Sehoon Kim**[1]     **Karttikeya Mangalam**[1]     **Suhong Moon**[1]
**Jitendra Malik**[1]     **Michael W. Mahoney**[123]     **Amir Gholami**[12]     **Kurt Keutzer**[1]

[1]University of California, Berkeley   [2]ICSI   [3]LBNL
{sehoonkim, mangalam, suhong.moon, malik, mahoneymw, amirgh, keutzer}@berkeley.edu

## Abstract

The recent emergence of Large Language Models based on the Transformer architecture has enabled dramatic advancements in the field of Natural Language Processing. However, these models have long inference latency, which limits their deployment and makes them prohibitively expensive for various real-time applications. The inference latency is further exacerbated by autoregressive generative tasks, as models need to run iteratively to generate tokens sequentially without leveraging token-level parallelization. To address this, we propose Big Little Decoder (BiLD), a framework that can improve inference efficiency and latency for a wide range of text generation applications. The BiLD framework contains two models with different sizes that collaboratively generate text. The small model runs autoregressively to generate text with a low inference cost, and the large model is only invoked occasionally to refine the small model's inaccurate predictions in a non-autoregressive manner. To coordinate the small and large models, BiLD introduces two simple yet effective policies: (1) the fallback policy that determines when to hand control over to the large model; and (2) the rollback policy that determines when the large model needs to correct the small model's inaccurate predictions. To evaluate our framework across different tasks and models, we apply BiLD to various text generation scenarios encompassing machine translation on IWSLT 2017 De-En and WMT 2014 De-En, and summarization on XSUM and CNN/DailyMail. On an NVIDIA T4 GPU, our framework achieves a speedup of up to $2.12\times$ speedup with minimal generation quality degradation. Furthermore, our framework is fully plug-and-play and can be applied without any modifications in the training process or model architecture. Our code is open-sourced[1].

## 1   Introduction

In recent years, the Transformer [63] has become the *de-facto* model architecture for a wide range of Natural Language Processing tasks. The potential of the Transformer architecture has been further enhanced by the emergence of Large Language Models (LLMs) with up to hundreds of billions of parameters trained on massive text corpora [2, 47, 50, 10, 23, 7, 55, 83, 62]. Despite their performance, efficiently running these models for inference is a challenge due to their large model size and runtime complexity. This limits their use in many applications that require real-time responses.

These computational inefficiencies are particularly pronounced in *autoregressive* generative tasks such as machine translation [3, 1], summarization [21], and language modeling [41]. For these tasks, models need to run iteratively to generate tokens sequentially, as each token is dependent on the previously generated tokens. This requires the models to load weight matrices, as well as the cached keys and values of previously generated tokens [46], for each token generation, thus preventing

---

[1]https://github.com/kssteven418/BigLittleDecoder

37th Conference on Neural Information Processing Systems (NeurIPS 2023).

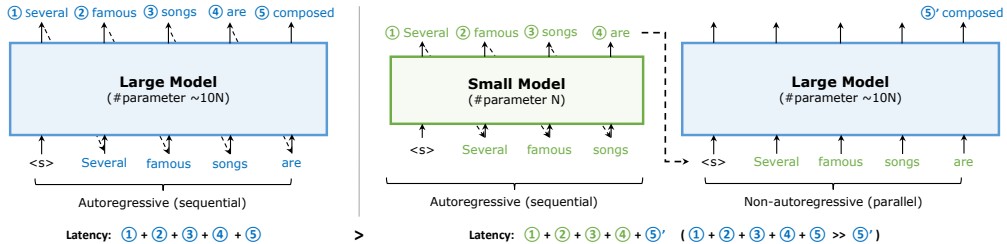

**Figure 1:** Illustration of (Left) the normal autoregressive decoding procedure of a large model and (Right) BiLD that consists of a small model and a large model. In BiLD, the small model generates tokens autoregressively (i.e., sequentially) until it hands over control to the large model. The large model then takes as input the tokens generated by the small model in parallel, allowing for non-autoregressive (i.e., parallel) execution to generate the next token. This improves end-to-end latency by allowing for more efficient utilization of underlying hardware.

parallelization of the loaded values across multiple tokens. This makes autoregressive text generation memory bandwidth constrained during inference [8]. As a consequence, autoregressive generative tasks suffer from low hardware utilization as well as high inference latency [32]. In contrast, non-autoregressive tasks, such as text classification [65], can process the entire input sequence with a single weight load, which is then shared across all input tokens in parallel. Given the increasing popularity of text generation tasks, in light of advancements in LLMs, it is critical to improve the inference latency and runtime efficiency of autoregressive decoding processes despite the potential sacrifice in generation quality.

To overcome this, non-autoregressive decoding [16, 69, 39, 59, 70, 52, 37, 15, 19] has been explored to maximize token-level parallelization and reduce the inference latency of generative tasks by generating multiple tokens simultaneously. This approach can be more computationally efficient than the regular autoregressive process. However, non-autoregressive decoding suffers from text generation quality issues due to its assumption of conditional independence between output tokens [29]. In order to achieve comparable performance to that of autoregressive processes, it generally requires complex and often task-dependent training strategies, supplementary hinting information that guides the decoding process [69, 39, 59, 70, 52], and knowledge distillation [84].

In this paper, we introduce a novel framework named Big Little Decoder (BiLD) that can be applied to various text generation scenarios to reduce inference latency *without* additional training iterations or modifications to the existing training pipeline or model architecture. As illustrated in Figure 1 (Right), the BiLD framework consists of two decoder models, a large model and small model, that work collaboratively to generate text sequences. In particular, only the small model is executed autoregressively to generate the majority of the text, taking advantage of its small runtime overhead. The large model only engages occasionally to refine the small model's inaccurate predictions, thus allowing for efficient non-autoregressive execution. This *autoregressive small, non-autoregressive large* scheme results in a substantial improvement of up to ~2× in end-to-end inference latency, compared to regular autoregressive execution, while maintaining similar or better generation quality. The effectiveness of our framework is also supported by our observation that the predictions made by small and large models only slightly disagree, and thus the small model can match the performance of the large model with a minimal refinement of its own predictions (Figure 2, Section 3.1).

In summary, our main contributions are as follows:

- We introduce BiLD, a general framework that allows faster inference of various text generation applications. Our framework is designed to coordinate a large model and a small model such that the large model is only executed infrequently and efficiently in a non-autoregressive manner to refine the small model's inaccurate predictions.

- We propose two policies for BiLD: the fallback policy that allows the small model to hand control over to the large model if it is not confident enough (Section 3.3), and the rollback policy that allows the large model to review and correct the small model's inaccurate predictions (Section 3.4).

- We introduce a simple yet effective technique for aligning the predictions of the small model with those of the large model. By incorporating this *prediction alignment* technique into the BiLD framework, we can further enhance its performance with minimal additional effort (Section 3.5.1).

- We apply BiLD for 4 different text generation scenarios including IWSLT 2017 De-En [3] and WMT 2014 De-En [1] for machine translation, XSUM [43] and CNN/DailyMail [21] for summarization. Compared to the full autoregressive execution, BiLD achieved a speedup of up to $1.85\times$ without generation quality degradation and $2.12\times$ allowing $\sim 1$ point degradation on an NVIDIA T4 GPU (Section 4.2).

## 2 Related Work

### 2.1 Efficient Transformer Decoding Inference

A variety of approaches have been proposed to increase the speed and reduce the overall inference costs of Transformers. Well-known approaches include efficient architecture design [26, 33, 36, 60, 67, 75], quantization [31, 54, 81, 82, 79, 74, 9], pruning [14, 49, 34, 42, 64, 12, 35], and neural architecture search [5, 56, 57, 66, 76, 80]. While these methods are generally suitable for Transformer-based tasks, some of the works have been focused on efficient decoding mechanisms to reduce the cost of autoregressive tasks.

One popular line of research that shares similarity to our work is *non-autoregressive decoding*. Non-autoregressive decoding, also known as parallel decoding, was first introduced in [16] as a method to reduce inference latency by producing multiple output tokens in parallel, thus avoiding sequential text generation. Subsequent work has further improved the performance of non-autoregressive models by incorporating auxiliary or hinting information [69, 39, 59, 70, 52] to ensure more accurate parallel decoding, or by allowing multiple additional iterations to refine any inaccurate predictions [37, 15, 19]. Such a multi-iteration decoding scheme has also been proposed in [71, 17, 58], which generates texts with fewer steps than the autoregressive scheme by inserting or deleting multiple tokens per iteration. However, these works require complex and often task-dependent training strategies and/or auxiliary information to achieve comparable performance to that of autoregressive models. In contrast, our methodology aims for a plug-and-play solution that does not require any complex training pipeline.

Our work is also related to the approaches that reduce the decoding cost by making decoders shallow. [29] demonstrates that increasing the depth of encoders and decreasing the depth of decoders can reduce decoding latency while still preserving performance. CALM [51] recently introduces *early exiting*, which dynamically adjusts the depth of the decoder for each token generation by terminating the inference at a middle layer, rather than executing until the end layer. While our method shares the same goal of accelerating decoding, we take a different approach by improving decoding parallelism rather than by skipping unnecessary computation. In addition, our framework offers several advantages over CALM: (1) our method is a fully black box approach that does not involve any modifications to model structures, while CALM requires modifications such as state propagation for the skipped layers; (2) our approach does not require changes to the training pipeline, whereas CALM requires averaged loss across all layers to ensure layer consistency; (3) our approach can be also applied without any training which is critical in various LLM use cases where training is either infeasible or prohibitively expensive. In Section 4.4, we also show that the early exiting strategy can be implemented in our framework to yield significantly better generation quality, further demonstrating the generalizability of our method to a wider range of problems.

### 2.2 Use of Multiple Models

Coordinating the use of multiple models has also been explored in knowledge distillation and ensemble learning. *Knowledge distillation* is a widely adopted methodology for enhancing the performance of smaller models by training them to replicate the behavior of larger, more complex models [22]. When applied to the Transformer architecture, this approach involves distilling the final logits [48, 61] and/or hidden states of a larger model, such as the attention map [60, 28, 68]. In contrast to knowledge distillation, which leverages the knowledge of a large model solely during the training time to improve the training of a smaller model, our method is a run-time solution applied during the decoding process. Therefore, our approach can be more adaptive to run-time behaviors and does not add complexity to training.

*Ensemble learning* is another approach for coordinating multiple models, wherein multiple models are trained independently and their predictions are combined to improve overall performance. Ensemble learning has been found to yield promising results for Transformer inference [44, 25, 77, 40, 24],

particularly when the models aggregated are pre-trained on different datasets and use different techniques. However, ensemble learning generally requires running multiple models and combining their predictions at run-time, which can be computationally expensive and not optimized for latency. Our research aims to optimize both model performance and run-time latency.

Concurrently and independently of our work, [38, 4] also propose an interesting algorithm to accelerate generative inference using a more powerful model to score and speculatively sample predictions from a less powerful model. While [38, 4] offer unbiased estimators that match the stronger model's probability distributions, our extensive empirical evaluation shows that our approach can deliver superior latency-performance trade-offs, due to its non-random rollback (i.e., rejection) policy as well as the dynamic fallback window size. See Section 4.2 and Appendix 7.3 for an in-depth comparison.

## 3 Methodology

### 3.1 Motivating Examples

Although large models tend to produce higher-quality text, they also result in longer end-to-end latencies, which can be further exacerbated by the regressive process of predicting one token at a time. However, in many text generation scenarios we demonstrate that a model that is an order of magnitude smaller than a larger model can achieve comparable generation quality to the larger model, provided that a few erroneous predictions are corrected. This implies that only a small fraction of the small model's predictions deviate from those of the larger model. To validate this claim, we evaluate two different generative scenarios, machine translation with mT5 [78] on WMT 2014 De-En [1] and summarization with T5 [47] on CNN/DailyMail [21] by running the large model along the small model for every decoding iteration. See Section 4.1 for more details on these models. Then, we measure the likelihood of the large model predicting the same token

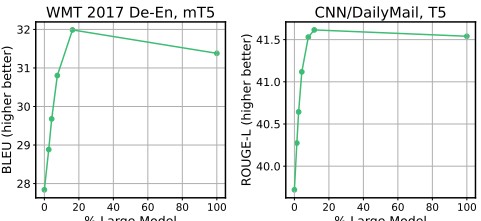

**Figure 2:** Quality of text generation for different proportions of the large model's engagement on the small model's prediction, evaluated on the validation datasets of (Left) WMT 2014 De-En translation [1]; and (Right) CNN/DailyMail summarization [21]. We see that the small models can achieve a comparable or better generation quality to the large models if ∼20% of their incorrect predictions were substituted.

that the small model generates. If the likelihood is below a certain threshold, we assume that the small model's prediction is not accurate enough, and we replace it with the large model's prediction. By controlling the threshold, we can adjust the proportion of the large model's engagement.

Figure 2 plots the text generation quality on the validation dataset of each benchmark for different proportions of the large model's engagement. The results exhibit a clear trend across the tasks where the small models with ∼10× smaller sizes can retain the large model's generation quality only if approximately 20% of their inaccurate predictions were substituted by the large model. While this experiment assumes an ideal case where the predictions of the large model are available as ground truth in every iteration, it nonetheless demonstrates the feasibility of achieving the text generation quality of the large model while maintaining the low inference latency of the small model.

### 3.2 Problem Formulation

At $n$th decoding iteration, the small model and the large model each take as input a partially generated output text $y_{1:n-1} = (y_1, \cdots, y_{n-1})$, and then generate a probability distribution over entire vocabulary $p_S(y|y_{1:n-1})$ and $p_L(y|y_{1:n-1})$, respectively. Then, the next token $y_{n,S}$ and $y_{n,L}$ are sampled from the probability distributions,

$$y_{n,S} \sim p_S(y|y_{1:n-1}) \quad \text{and} \quad y_{n,L} \sim p_L(y|y_{1:n-1}). \tag{1}$$

Depending on whether to use the small model or the large model for the $n$th decoding step, the $n$th token $y_n$ can be either $y_{n,S}$ or $y_{n,L}$. When deciding which model to use, it is not feasible to run the large model along with the small model for every decoding step to verify the predictions of the small model, as in the experiments in Section 3.1. Thus, it is necessary to hand over the control to the

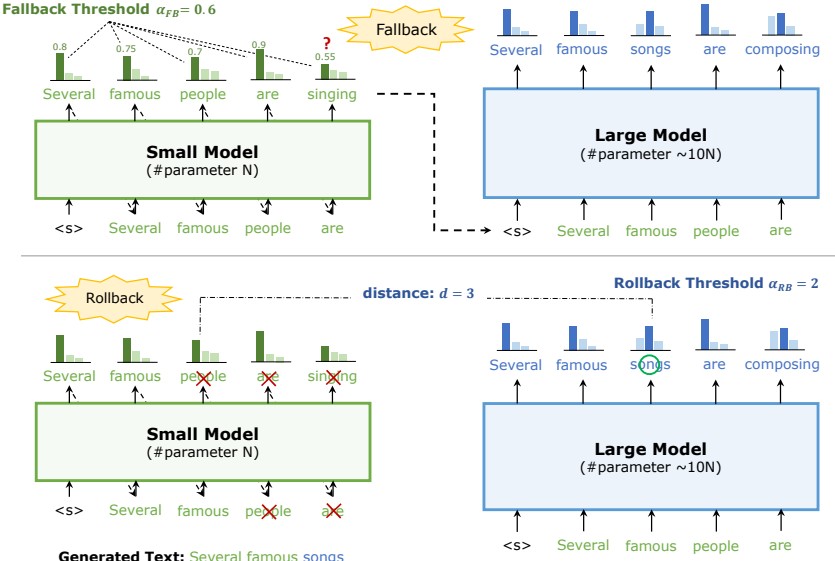

**Figure 3:** (Top) The fallback policy. When the small model generates tokens autoregressively, if the prediction probability of a specific token is below the predefined fallback threshold value $\alpha_{FB}$, the prediction is deemed to be not confident enough, and control is then shifted to the larger model to produce the corresponding token. (Bottom) The rollback policy. If the large model takes over the control, it produces its own predictions for all previous tokens, as well as the current token. If the prediction probability from the large model for a previously generated token deviates from the small model's prediction by a distance metric $d$ exceeding the predetermined rollback threshold $\alpha_{RB}$, the small model's prediction is regarded as incorrect. In such a case, we roll back all predictions made by the small model that follow the corresponding token.

large model only when the small model is likely to make an inaccurate prediction based on a policy $\pi(y_{1:n-1})$ that returns a boolean value $\{0, 1\}$ indicating whether to use the large model:

$$y_n = \begin{cases} y_{n,S} & \text{if } \pi(y_{1:n-1}) = 0 \\ y_{n,L} & \text{if } \pi(y_{1:n-1}) = 1. \end{cases} \quad (2)$$

The objective, therefore, is to design a lightweight policy $\pi$ that leads to high text generation quality with the minimum end-to-end latency by invoking the large model only when necessary. In order to illustrate the mechanism by which latency is reduced, consider a simple case where the small model has generated tokens $y_1$ through $y_n$ autoregressively. If the large model takes over the control and predicts the next token, $y_{n+1}$, it can now take multiple input tokens ($y_1$ through $y_n$) in parallel, thus allowing for *non-autoregressive* inference. It is worth noting that this non-autoregressive approach would require the same amount of FLOPs as a regressive approach that predicts $y_1$ through $y_{n+1}$ sequentially; however, it is much faster on hardware due to its token-level parallelism and increased arithmetic intensity [72]. In other words, processing multiple tokens in a single memory operation is more efficient than individually processing tokens with separate memory operations, as memory accesses can be more costly than arithmetic operations in decoding tasks [32]. If the latency saving from running the large model non-autoregressively outweights the additional cost of running the small model, there is a net latency reduction. Therefore, the aim of this approach is *not* to reduce the number of FLOPs, but rather to improve the hardware utilization and arithmetic intensity of the decoding process. More detailed analysis on this can be found in Appendix 7.5.1. Figure 1 provides a high-level overview of how the small and the large models in BiLD coordinates for text generation.

We now focus on constructing an appropriate policy $\pi$ for our framework. Here, we introduce two simple policies, the fallback and rollback policies, which (despite their simplicity) result in high performance with significant latency reduction. We discuss the details in the following subsections.

### 3.3 Fallback Policy: Small Model Knows When to Stop Predictions

The first principle of the policy is that the small model should be able to determine when to hand over control to the large model. Whenever the small model lacks confidence in its prediction, it is

better to allow the large model to take over. Confidence (or uncertainty, in reverse) quantification has been an active research area [13, 30], and any lightweight confidence metric can serve as a potential candidate. Here, we find it sufficient with a simple policy based on the maximum prediction probability, i.e., $\max_y p_S(y|y_{1:n-1})$, similar to the observations made in [20]. If the maximum prediction probability is lower than a certain threshold $\alpha_{FB}$, then the small model's prediction is regarded to be not confident enough, and we *fallback* to the large model to generate the next token. Note that this does not entail a runtime overhead. Figure 3 (Top) illustrates the fallback policy.

**Fallback Policy:** If $\max_y p_S(y|y_{1:n-1}) < \alpha_{FB}$, then fallback to the large model and set $y_n = y_{n,L}$.

## 3.4 Rollback Policy: Large Model Knows When to Revert Predictions

While the fallback policy allows the large model to take over when the small model is not confident enough, it is still possible that the small model is over-confident in its incorrect predictions [18]. Moreover, a single incorrect prediction at an early decoding iteration can lead to a catastrophic effect [51], as it will affect all subsequent token predictions. To avoid such cases, it is desirable to have the large model review the small model's predictions and ensure the validity of each prediction. In our framework, this comes without any extra cost. When the large model is provided with the tokens generated by the small model for its non-autoregressive prediction of the next token, it also produces its own predictions for all the previous decoding steps. That said, given the partially generated text $y_{1:n}$, it generates $p_L(y|y_{1:m})$ for all previous and current decoding steps $m = 1, \cdots, n$, which can be used to validate the small model's previous predictions.

Therefore, for some distance metric $d(\cdot, \cdot)$ that compares two probability distributions, we find the smallest decoding step $m$ such that

$$d(p_S(y|y_{1:m}), p_L(y|y_{1:m})) > \alpha_{RB} \tag{3}$$

for a predetermined threshold $\alpha_{RB}$. If such $m$ exists, we regard the small model's previous prediction $y_m$ to be inaccurate, and we *rollback* all predictions that follow, i.e., $y_m$ through $y_n$, since they are all dependent on the wrong prediction. We then replace $y_m$ with $y_{m,L}$ of the large model. We will discuss in Section 4.2 that the cross-entropy loss between the small model's hard label and the large model's soft label (which measures the likelihood of obtaining the small model's prediction from the large model's output) is a good choice for the metric $d$. Rollback may incur additional latency due to the need for duplicated computation for the reverted tokens. However, we demonstrate in Section 4.3 the net advantage of rollback as the improved text generation quality outweighs the additional latency. See Figure 3 (Bottom) for a detailed illustration of the rollback policy.

**Rollback Policy:** If there exists a minimum $m \in [1, n-1]$ such that $d(p_S(y|y_{1:m}), p_L(y|y_{1:m})) > \alpha_{RB}$, then rollback the predictions $(y_m, \cdots, y_n)$ and set $y_m = y_{m,L}$.

## 3.5 Big Little Decoder

Taken together, the Big Little Decoder (BiLD) framework consists of one small model, one large model, and a policy that determines which model to use for each decoding iteration. The policy comprises two components: the fallback policy to fall back to the large model when the small model's prediction is not confident enough; and a rollback policy to roll back the small model's predictions if they deviate from the predictions of the large model. Algorithm 1 provides a summary of the end-to-end algorithm.

**Algorithm 1:** Big Little Decoder

```
 1: y ← []
 2: while y[−1] ≠ <eos>
 3:     p_S ← SmallModel(y)
 4:     if max(p_S[−1]) > α_FB
 5:         # Use the small model's predicton
 6:         y ← y + [sample(p_S[−1])]
 7:     else
 8:         # Fallback to the large model
 9:         p_L ← LargeModel(y)
10:         m ← min. index such that d(p_L[m], p_S[m]) > α_RB
11:         if m exists
12:             # Rollback: use the large model's prediction
13:             y ← y[: m] + [sample(p_L[m])]
14:         else
15:             # Don't rollback: use the large model's prediction
16:             y ← y + [sample(p_L[−1])]
17: return y
```

### 3.5.1 Model Prediction Alignment

BiLD is a general framework that imposes no restriction on the selection of small and large models as long as they use the same vocabulary. Therefore, as will be demonstrated in Section 4.2, two

independently trained models can compose BiLD to achieve a significant latency improvement. Nevertheless, when two models are trained separately, they may generate sequences with similar or identical semantic meanings but using different vocabularies. For instance, one model may produce the phrase "writing is hard" while the other may generate "writing is difficult". Because the BiLD policy relies on the degree of agreement between the large and small models, such a vocabulary-level discrepancy can result in unnecessary disagreements that roll back the small model's prediction without any improvement in generation quality.

In order to address this issue and further optimize the BiLD performance, we present a simple approach called *model prediction alignment* that aids in aligning the predictions produced by the small and large models. To achieve this, we leverage a calibration dataset $\mathcal{X}_{\text{cal}} = \{x^{(i)}\}$ that well represents the input sentence distribution. We then generate the corresponding output sequence for each input sequence using the large model, resulting in $\mathcal{Y}_{\text{cal}} = \{y^{(i)}\}$ where $y^{(i)} = \arg\max p_L(y|x^{(i)})$. Subsequently, we fine-tune the small model using the calibration examples $(x_{\text{cal}}, y_{\text{cal}}) \in (\mathcal{X}_{\text{cal}}, \mathcal{Y}_{\text{cal}})$.

The underlying rationale of this approach is to increase the likelihood of the small model generating sequences that would have been generated by the large model. This can minimize the distance between the small model and the large model's predictions per each token, i.e., $d(p_S(y|x, y_{1:m}), p_L(y|x, y_{1:m}))$, throughout the decoding process, thereby avoiding unnecessary rollbacks. Despite its simplicity, our experiments in Section 4.2 demonstrate that this approach can be incorporated into the BiLD framework with minimal effort to significantly enhance the performance. We further emphasize that this method does not introduce any additional complexity or hyperparameters to the normal training pipeline. This is comparable to knowledge distillation [22], an alternative method for aligning model predictions, which requires modifications to the training pipeline, access to internal hidden states such as logits, and additional hyperparameter tuning.

# 4 Evaluations

## 4.1 Experiment Setup

**Models and Datasets.** To access the generalizability and validity of BiLD in various text generation settings, we have selected IWSLT 2017 De-En [3] and WMT 2014 De-En [1] for machine translation benchmarks and XSUM [43] and CNN/DailyMail [21] for summarization benchmarks. We used mT5-large and small [78] for machine translation and T5-large and small [47] for summarization as our target models, where the size of the models differ by approximately a factor of 20. Our framework is built on top of PyTorch [45] and the HuggingFace Transformers library [73] along with their pre-trained checkpoints.

**Training.** We fine-tune the pre-trained models on the target benchmarks for 500k steps to obtain the *baseline* small and large models. To train the *aligned* small models via the prediction alignment method (Section 3.5.1), we generate output sequences from the input sequences of the training datasets using the fully trained large models to create calibration datasets. We then fine-tune the pre-trained small models on the calibration dataset using the same training recipes and the number of steps as the baseline small models. More training details can be found in Appendix 7.1.1. Throughout the paper, we refer to BiLD with the baseline and aligned small models as *unaligned* and *aligned* BiLD, respectively.

**Inference.** All inference evaluations are conducted on a single NVIDIA T4 GPU of a GCP n1-standard-4 instance, using a batch size 1, which is a common use case for online serving [51]. For the distance metric $d$ in Equation 3 for the rollback policy, we use the cross-entropy loss between the small model's hard label and the large model's soft label. For BiLD inference, we sweep over different fallback and rollback thresholds to explore different trade-offs between generation quality and latency. More evaluation details can be found in Appendix 7.1.2.

## 4.2 Main Results

The main results are illustrated in Figure 4, which shows the trade-off between text generation quality and average end-to-end latency per example, normalized by the vanilla inference latency of the pure large baseline models. The trade-offs are obtained by controlling the fallback and rollback thresholds. Table 1 summarizes the results, with the second and third rows corresponding to unaligned

**Table 1:** The summary of Figure 4 which compares the generation quality and latency speedup of BiLD against vanilla inference with large baseline models. The first row reports the vanilla inference, and the second and third rows report unaligned BiLD. The fourth and fifth rows report aligned BiLD. In both cases of unaligned and aligned BiLD, we report the speedup with minimal BLEU/ROUGE-L score degradation (second and fourth rows), and within ∼1 point degradation (third and fifth rows).

| Task (Model) | Machine Translation (mT5) | | | | Summarization (T5) | | | |
|---|---|---|---|---|---|---|---|---|
| Dataset | IWSLT | | WMT | | XSUM | | CNN/DailyMail | |
| | BLEU | Speedup | BLEU | Speedup | ROUGE-L | Speedup | ROUGE-L | Speedup |
| Vanilla Inference | 40.32 | - | 31.38 | - | 35.08 | - | 41.54 | - |
| BiLD (Unaligned) | 40.33 | 1.43× | 31.28 | 1.34× | 35.12 | 1.48× | 41.44 | 1.71× |
| | 39.44 | 1.58× | 30.47 | 1.43× | 34.02 | 1.72× | 40.57 | 2.05× |
| BiLD (Aligned) | 40.24 | 1.62× | 31.26 | 1.47× | 35.05 | 1.50× | 41.52 | 1.85× |
| | 39.13 | 1.78× | 30.33 | 1.70× | 33.95 | 1.80× | 40.96 | 2.12× |

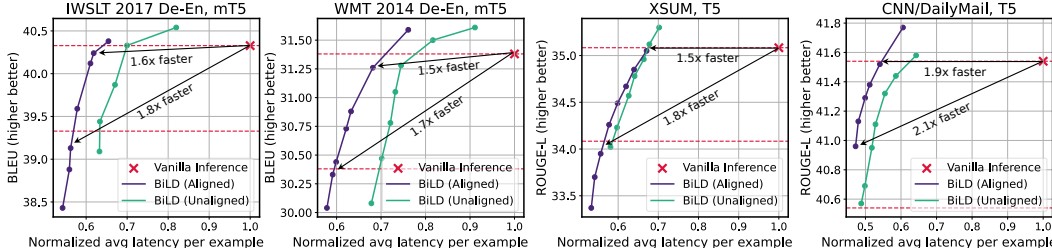

**Figure 4:** Generation quality and average end-to-end latency of processing a single example on 4 different benchmarks. We report BLEU for machine translation and ROUGE-L for summarization as performance metrics. The green and blue lines are unaligned and aligned BiLD, respectively. The X marks are the vanilla inference with the baseline large models. For comparison, two horizontal lines are plotted to indicate the BLEU/ROUGE-L score of the vanilla inference and 1 point degradation from it. The latency on the x-axis is normalized by the baseline latency.

BiLD. When coupled with the normally fine-tuned baseline small models, BiLD achieves an average speedup of 1.50× across all benchmarks, with up to 1.71× speedup on CNN/DailyMail without any degradation in text generation quality (2nd row). By allowing ∼1 point degradation, BiLD achieves an average speedup of 1.70×, with up to 2.05× speedup (3rd row). Note that unaligned BiLD is a *pure* plug-and-play solution that does not require additional training effort or cost beyond preparing small and large models independently.

In addition, Figure 4 shows the efficacy of the prediction alignment method, leading to a consistent improvement of aligned BiLD over unaligned BiLD. As summarized in the forth and fifth rows of Table 1, aligned BiLD that incorporates the aligned small models yields an average speedup of 1.61×, with up to 1.85× speedup (4th row). Within ∼1 point degradation, it achieves an average speedup of 1.85×, with up to 2.12× speedup (5th row). The results also demonstrate that both unaligned and aligned BiLD outperform the baseline BLEU/ROUGE-L scores in the high-latency regime, which can be attributed to the ensembling effect of using two different models, as also studied in prior work [40]. In Appendix 7.5.2, we provide examples of text sequences generated by BiLD, which demonstrate that the large model's engagement in BiLD decoding not only improves the prediction accuracy but also prevents incorrect predictions from impacting the future ones.

We have additionally conducted a performance comparison of our method with the speculative sampling method proposed in [4] on the IWSLT 2017 De-En and XSUM benchmarks. We implement and evaluate it in the same environment as our main BiLD experiments using the same baseline large and small models. We apply a fixed window size of [3, 10]. On the IWSLT benchmark, speculative sampling achieves a BLEU score of 39.93 with a 1.28× speedup, while BiLD (unaligned) achieves a 0.61 higher BLEU score with similar speedup, or a 0.21× more latency gain with a similar BLEU score. On the XSUM benchmark, speculative sampling achieves a ROUGE-L score of 35.00 with a 1.25× speedup. In contrast, BiLD achieves up to a 0.30 ROUGE-L score gain with a faster latency, or up to 0.22× more latency gain with a better ROUGE-L score. We provide more detailed comparisons in Appendix 7.3.

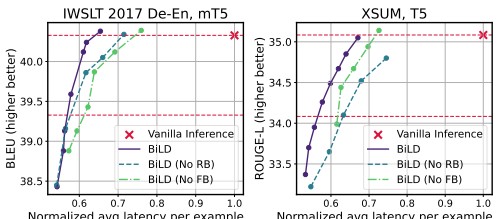

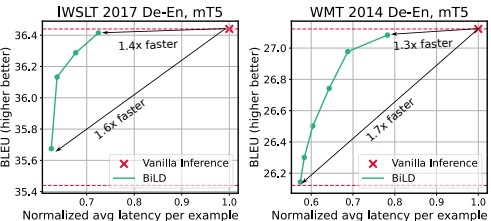

**Figure 5:** Ablation study results for BiLD on (Left) IWSLT 2017 De-En translation and (Right) XSUM summarization tasks without the rollback or fallback policy. Aligned small models were used in all cases. The result demonstrates that BiLD experiences significant performance degradation without either policy in both tasks. The horizontal lines indicate the vanilla inference score and 1 point degradation from it.

**Figure 6:** Application of the BiLD framework to the early exit problem using the mT5-small model as the large model and its first layer as the small model, evaluated on (Left) the IWSLT 2017 De-En and (Right) WMT 2014 De-En benchmarks. The × marks indicate the latency and BLEU score of the mT5-small models. The horizontal lines indicate the vanilla inference score and 1 point degradation from it.

### 4.3 Ablation Studies

We have further conducted two ablation studies to validate the individual components of BiLD by (1) removing the rollback policy, and (2) removing the fallback policy. When removing the rollback policy, we use the same fallback thresholds as the main experiments to control the generation quality and latency trade-off. When removing the fallback policy, we use the same rollback thresholds as the main experiments. Additionally, we apply fallback after a fixed number of small model executions (swept over [3, 10]), similar to [4].

Figure 5 illustrates the results of these ablation studies on IWSLT 2017 De-En for machine translation and XSUM for summarization with aligned BiLD. The results show that the rollback policy consistently produces better generation quality across all latency regimes, particularly in the high-BLEU/ROUGE regime where the large model's engagement via rollback is crucial in correcting small model's wrong predictions. This demonstrates that, despite the additional latency overhead from the duplicated computation of reverted tokens, the improvement in text generation quality outweighs this cost. Similarly, removing the fallback policy and periodically handing over control to the large model after a fixed number of token generations leads to significant performance degradation. Taken together, these results highlight that both policies are critical components of BiLD.

### 4.4 Early Exiting Strategy in the BiLD Framework

So far, we have demonstrated how BiLD can be used as a general framework for accelerating the text generation process by incorporating a small model and a large model. However, having two separate models is not a limitation as they can be combined into a single model by using a subset of a larger model, such as a few of its early layers, as a smaller model. This approach resembles the early exit strategy, which is a popular method for accelerating the decoding process [51]. This section demonstrates how the early exiting strategy can be reframed within the BiLD framework.

To demonstrate the applicability of using the early exiting strategy within the BiLD framework, we use mT5-small model as the large model and the first (out of 8) layer as the small model, and evaluate it on two machine translation benchmarks: IWSLT 2017 De-En and WMT 2014 De-En. To ensure consistency between the prediction made after the first layer and the one made after the last layer, we train the model with the average loss of these two layers, similar to [11, 51]. The prediction head is shared for these two layers. More training and evaluation details can be found in Appendix 7.2.1.

Figure 6 illustrates the results, where for each benchmark, BiLD achieves up to 1.60× and 1.74× speedup within less than one point BLEU score drop, respectively. This demonstrates the extensibility of the BiLD framework to early exit problems. In Appendix 7.2.2, we further provide a detailed comparison of our results with CALM [51], another framework that incorporates early exiting for fast Transformer decoding. Compared to CALM, BiLD offers two advantages that contribute to better generation quality: (1) in BiLD, even if an early exited prediction (i.e., prediction made by the smaller model) is incorrect, it can be corrected and replaced using the rollback policy; (2) the key and value caches for skipped layers are filled with actual values instead of being computed from the

exiting layer's hidden states, leading to reduced error propagation and improved decoding stability. As a result, when tested on IWSLT 2017 De-En and WMT 2014 De-En using mT5-small, BiLD achieves a BLEU score improvement of up to ∼2 points over CALM in both datasets (Figure 7).

## 5    Conclusion

In this work, we have introduced Big Little Decoder (BiLD), a framework that reduces end-to-end inference latency for a wide variety of text generation tasks without the need for training or modifying the existing models. In essence, our framework couples a large and small decoder model together to generate text more efficiently. In particular, we start inference with a small model which runs autoregressively for the majority of the time to generate text with a low inference cost, while the large model is executed non-autoregressively to refine the small model's inaccurate predictions. BiLD incorporates two policies, the fallback policy, which hands control to the large model when the small model is uncertain, and the rollback policy, which allows the large model to revert the small model's inaccurate predictions. Our framework is evaluated across various text generation scenarios, including machine translation, summarization, and language modeling. Running on an NVIDIA Titan Xp GPU, with no performance drop BiLD achieved an average speedup of $1.52\times$, with improvements of up to $2.18\times$ on some tasks. Furthermore, when a 1 point degradation in performance was allowed, BiLD achieved an average speedup of $1.76\times$ with speedups of up to $2.38\times$ on some tasks.

## 6    Acknowledgements

We acknowledge gracious support from Google Cloud, Google TRC team, and specifically Jonathan Caton, Prof. David Patterson, and Dr. Ed Chi. Prof. Keutzer's lab is sponsored by Intel corporation, Intel VLAB team, Intel One-API center of excellence, as well as funding through BDD and BAIR. Sehoon Kim and Suhong Moon would like to acknowledge the support from Korea Foundation for Advanced Studies (KFAS). Amir Gholami was supported through funding from Samsung SAIT. Michael W. Mahoney would also like to acknowledge a J. P. Morgan Chase Faculty Research Award as well as the DOE, NSF, and ONR. Our conclusions do not necessarily reflect the position or the policy of our sponsors, and no official endorsement should be inferred.

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

# 7 Supplementary Material

## 7.1 Experimental Details

**Table 2:** Model configurations of the large and small models for each evaluation task. For comparison, the number of layers, hidden dimension, FFN dimension, and the number of decoder parameters (without embeddings) for each model are provided.

| Task | Model | # Layers | dim | FFN dim | # Params |
|---|---|---|---|---|---|
| Machine Translation | mT5-large [78] | 24 | 1024 | 2816 | 409M |
| | mT5-small [78] | 8 | 512 | 1024 | 25M |
| Summarization | T5-large [47] | 24 | 1024 | 4096 | 402M |
| | T5-small [47] | 6 | 512 | 2048 | 25M |

### 7.1.1 Training Details

For machine translation, we use IWSLT 2017 German-English [3] and WMT 2014 German-English [1] as target benchmarks, and mT5 [78] as a target model. We use the 8-layer mT5-small and the 24-layer mT5-large as the small and large models. For summarization, we use XSUM [43] and CNN/DailyMail [21] as target benchmarks, and T5 [47] as a target model. We use T5-small and T5-large with 6 and 24 layers, respectively, for the small and large models. Table 2 summarizes the size and configuration of each model. All the models are fine-tuned from the pre-trained checkpoints of the HuggingFace library [73] for 500k steps using a batch size of 16. We use Adafactor optimizer [53] with constant learning rate of $\{0.5, 1, 2, 5\}e-4$ for the small models and $\{0.5, 1\}e-4$ for the large models. We refer to the normally fine-tuned models on the validation datasets as the *baseline* small and large models.

When training *aligned* small models via the prediction alignment method described in Section 3.5.1, we first generate calibration datasets using the input sequences from the training datasets of each benchmark. We then use the fully trained large model to generate output sequences through greedy sampling with a beam size of 1. To ensure a fair comparison, we fine-tune pre-trained small models (rather than the baseline small models that are already fine-tuned on the training datasets) on the calibration datasets using the same training recipes and the number of training steps as described above. This decision is based on our observation that fine-tuning a baseline model using the calibration dataset tends to improve generation quality, likely due to the increased number of training examples and data augmentation effects, which makes it difficult to make a fair comparison between unaligned BiLD and aligned BiLD. However, in practice, one can obtain aligned models by applying the prediction alignment method directly to the fine-tuned baseline small models to achieve the best performance.

### 7.1.2 Evaluation Details

All inference evaluations including latency measurement are conducted on a single NVIDIA T4 GPU of a GCP n1-standard-4 instance with 4 vCPUs and 15GB memory. For inference, we use batch size 1, which is a common use case for online serving [51]. For the distance metric $d$ in Equation 3 for the rollback policy, we use the cross-entropy loss between the small model's hard label and the large model's soft label. This measures the (negative log) likelihood of obtaining the small model's prediction from the large model's output. For BiLD inference, we sweep over different fallback and rollback thresholds to explore different trade-offs between generation quality and latency. For the machine translation tasks, we use fallback thresholds in [0.5, 0.9] and rollback thresholds in [1, 10]. For the summarization tasks, fallback thresholds in [0.2, 0.6] and rollback thresholds in [2, 6]. We keep the maximum generation length of the small model to 10 to avoid high rollback costs. In Appendix 7.5.3, we provide a detailed analysis of how varying the fallback and rollback thresholds impacts the trade-offs between generation quality and latency in the BiLD framework.

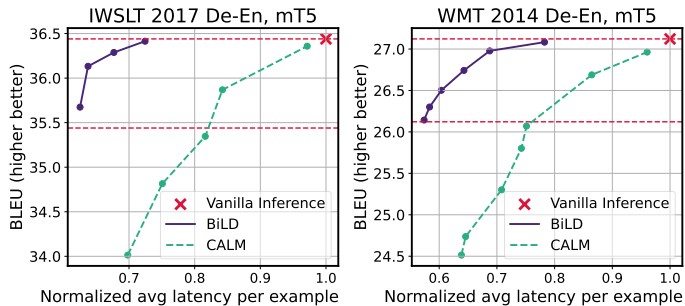

**Figure 7:** The trade-off curves between inference latency and BLEU score for BiLD and CALM in the early exiting setting for (Left) IWSLT 2017 De-En and (Right) WMT 2014 De-En. The × marks indicate the vanilla inference latency and BLEU score of the mT5-small models. The horizontal lines indicate the vanilla inference score and 1 point degradation from it. BiLD outperforms CALM across all speedup regimes by up to $2 \sim 2.5$ points better BLEU score, demonstrating the effectiveness of our approach for the early exiting strategy.

## 7.2 Details of Early Exiting Strategy in the BiLD Framework

### 7.2.1 Training and Evaluation Details

The training and evaluation details for BiLD as well as for CALM are as follows.

**BiLD.** We use the mT5-small model as the large model and the first (out of 8) layer as the small model, and evaluate it on two machine translation benchmarks: IWSLT 2017 De-En and WMT 2014 De-En. To ensure consistency between the prediction made after the first layer and the one made after the last layer, we fine-tune the pre-trained mT5 model using the average loss of the first and the final layers, similar to [11, 51]. That is, $\mathcal{L} = \frac{1}{2}(\mathcal{L}_1 + \mathcal{L}_{-1})$ where $\mathcal{L}_1$ and $\mathcal{L}_{-1}$ are the negative log-likelihood loss after the first layer and the final layer. The prediction head is shared for these two layers. We fine the pre-trained mT5-small model on each benchmark for 500k steps using a batch size of 16. Similar to the main experiments, we use Adafactor optimizer [53] with constant learning rate of $\{0.5, 1, 2, 5\}\mathrm{e}{-}4$. For evaluation, we use fallback thresholds in [0.2, 0.8] and rollback thresholds in [0.5, 1.5].

**CALM.** To reproduce CALM [51] in our experimental setup, we have fine-tuned the pre-trained mT5-small model on IWSLT 2017 De-En and WMT 2014 De-En datasets. We employ the averaged loss across all layers, i.e., $\mathcal{L} = \sum_{i=1}^{L} w_i \mathcal{L}i$, where $w_i = i / \sum_{j=1}^{L} j$, which was introduced in the paper to ensure the layer consistency. We use Adafactor optimizer [53] with constant learning rate of $\{0.5, 1, 2, 5\}\mathrm{e}{-}4$ for 500k training steps. To make a fair comparison, we match the BLEU score of the fine-tuned model to that of BiLD's models Among the two training-free confidence measures introduced in the CALM paper, softmax-based and hidden-state saturation-based measures, we have chosen to use the latter approach as an early exiting criterion. That said, if the cosine similarity between the current layer's hidden states and the previous layer's hidden states exceeds a certain threshold, we perform early exiting. We have found that the softmax-based alternative is not applicable in our evaluation scenario due to the large output vocabulary (more than 200k for mT5, which is $\sim 10\times$ larger than T5), which significantly increases latency overhead. As described in the paper, when early exiting happens, the hidden states of the exited layer are propagated down to the remaining layers to compute the key and value caches. To achieve different trade-offs between latency and generation quality, we sweep over $\lambda$ in [0.7, 0.98] and $t$ in $\{0, 1, 2, 4, 8\}$ in the decaying threshold function.

### 7.2.2 Performance Comparison between BiLD and CALM

Figure 7 illustrates the BLEU score and latency curves of BiLD compared to CALM in the early exiting setting. In both tasks, our method achieves significantly better BLEU scores with the same latency speedup, yielding up to around 2 point better BLEU score in the $\sim 1.5\times$ speedup regime. This can be attributed to two factors. First, in BiLD, even if an early exited prediction (i.e., prediction made by the smaller model) is incorrect, it can be corrected and replaced using the rollback policy.

Therefore, an error in the early exited layer is propagated less drastically to the future prediction. Second, the key and value caches for skipped layers are filled with actual values instead of being computed from the exiting layer's hidden states. This also leads to reduced error propagation and improved decoding stability.

## 7.3 Comparison with Other Speculative Decoding Frameworks

Concurrently and independently of our work, [38, 4] also propose an algorithm to accelerate generative inference using a more powerful model to score and speculatively sample predictions from a less powerful model. While the rejection sampling-based approach in [38, 4] offers unbiased estimators that match the stronger model's probability distributions, our extensive empirical evaluation shows that our approach can deliver superior latency-performance trade-offs, due to its non-random rollback (i.e., rejection) policy as well as the dynamic fallback window size. Below, we provide distinctions in detailed methodologies and quantitative comparison, as well as our insights on better latency and performance of our approach.

### 7.3.1 Differences in methodology

While the idea of using two models with different sizes can be deemed similar to the speculative decoding frameworks in [38, 4], we have clear distinctions in detailed methodologies.

**(1) Non-Random Prediction Rollback Approach:** The primary difference lies in how we decide the rollback (e.g., rejection) of predictions from the small model. In our rollback policy, we propose to make the rejection decision based on the *distance* between the small and large model predictions, which differs from the rejection sampling policy outlined in [38, 4]. While [38, 4] propose an unbiased estimator on the large model's prediction, Figure 2 demonstrates that combining predictions from both models through our distance-based rejection approach can surpass the exclusive utilization of the large model's prediction probability. BiLD seeks to find and utilize this optimal performance point without introducing much runtime cost. We have a further discussion below about how our rejection policy benefits text-generation performance.

**(2) Dynamic Fallback Window Size:** Additionally, we introduce the *dynamic fallback window size* in our fallback policy. In [38, 4], the window size remains a fixed hyperparameter; however, it is also highlighted in [4] that the window size can have a noticeable impact on end-to-end latency. Our approach offers an efficient and robust solution: adjusting the window size at runtime based on the small model's confidence level in run-time. Our ablation study (Figure 5) demonstrates that omitting the fallback policy and periodically transitioning control to the large model, as proposed in [38, 4], can result in notable latency degradation.

**(3) Model Alignment Enhancement:** Beyond the core framework, we introduce a model alignment method to align the small model's predictions with those of the large model. This enhances the framework by reducing unnecessary rejections and can be incorporated with minimal adjustments to the training pipeline.

### 7.3.2 Quantitative Comparisons

In Table 3, we provide a comprehensive quantitative comparison between our method and [38, 4] across two different datasets: IWSLT for machine translation and XSum for summarization. In order to ensure a fair comparison that isolates the impact of the frameworks themselves, we employ the baseline (non-aligned) small model for all experiments. We maintained the same evaluation setup and hyperparameter space that are outlined in Appendix 7.1.2.

Table 3 includes two BiLD configurations: the one that matches latency and the other that matches BLEU/ROUGE-L scores as compared to the rejection sampling-based methods. Across all experiments, BiLD consistently outperforms speculative decoding. It achieves either (1) notably improved BLEU/ROUGE-L scores with equivalent latency gains, or (2) superior latency gains while retaining the same BLEU/ROUGE-L scores.

**Table 3:** Comparison of BiLD to other rejection sampling based speculative sampling methods proposed in [38, 4] on IWSLT and XSUM. For BiLD, we include two BiLD configurations: the one that matches latency and the other that matches BLEU/ROUGE-L scores as compared to the rejection sampling based methods. Note that BiLD consistently outperforms other methods by achieving either (1) improved BLEU/ROUGE-L scores with equivalent latency gains, or (2) improved latency gains while retaining the same performance score.

| Dataset | IWSLT | | XSUM | |
|---|---|---|---|---|
| | BLEU | Speedup | ROUGE-L | Speedup |
| Vanilla Inference | 40.32 | - | 35.08 | - |
| Rejection Sampling Based [38, 4] | 39.93 | 1.28× | 35.00 | 1.25× |
| BiLD (Match Latency) | **40.54** | 1.23× | **35.30** | 1.42× |
| BiLD (Match BLEU/ROUGE-L) | 39.87 | **1.49×** | 34.96 | **1.50×** |

**Table 4:** Comparison of the percentage of fallback and rollback (rejection) occurrences of BiLD and other rejection sampling based speculative sampling methods [38, 4]. While achieving even better BLEU/ROUGE-L scores in IWSLT and XSUM, BiLD involves noticeably fewer number of fallbacks and rollbacks, resulting in a significantly better latency speedup.

| Task | Method | BLEU/ROUGE-L | Speedup | % Fallback | % Rollback (rejection) |
|---|---|---|---|---|---|
| IWSLT | Rejection Sampling Based [38, 6] | 39.93 | 1.28× | 23.24% | 9.81% |
| | BiLD (Better BLEU) | 40.33 | 1.43× | **21.09%** | **1.56%** |
| XSUM | Rejection Sampling Based [38, 6] | 35.00 | 1.25× | 36.84% | 24.24% |
| | BiLD (Better ROUGE-L) | 35.12 | 1.48× | **32.33%** | **6.41%** |

### 7.3.3 Insights on Better Latency and Performance

Quantitative analysis reveals a consistent trend where our method, when compared to other speculative decoding frameworks, effectively enhances both text generation quality and latency. We provide insights and explanations into why our approach surpasses speculative decoding frameworks.

**(1) Better text generation quality**

**Ensembling effect:** The power of blending outputs from multiple models has been well-explored in various fields. This is also the case in open-source LLM models which exhibit diverse strengths and weaknesses due to variations in data, architectures, and hyperparameters, making different models complementary to each other [27]. In fact, we show such effects of blending multiple model outputs in Figure 2, where a combination of 20% of the large model's prediction with the small model's prediction outperforms the exact imitation of the large model's behavior. Our approach offers fallback and rollback policies that efficiently exploit optimal ensemble point, which produces superior output quality compared to the unbiased estimate of the large model in [38, 4].

**Rollback policy that leads to higher performance:** Our method adopts a rejection policy that completely discards the small model's prediction if it significantly deviates from the large model's counterpart, based on cross entropy-based distance metric. This contrasts with speculative decoding, where the decision involves a stochastic rejection sampling process. We empirically observe that BiLD 's *hard rejection policy* allows a better BLEU/ROUGE-L score with significantly fewer number of rollbacks (rejections) than the *stocastic rejection policy* of speculative decoding as described in Table 4. We hypothesize that this boost in predictive performance stems from our hard rollback policy, which prevents potentially erroneous predictions by ruling out stochasticity. We additionally hypothesize that such a strategy can address exposure bias, mitigating the impact of a single early-stage misprediction on subsequent predictions.

**(2) Lower end-to-end latency** Furthermore, our fallback policy introduces a dynamic fallback window size (i.e. number of small model's consecutive iterations) that is determined based on the run-time prediction confidence of the small model. This is in contrast with speculative decoding which adopts a static window size. The advantages of the dynamic window size are two-fold:

**Table 5:** BiLD with nucleus sampling (p=0.8) on IWSLT and XSUM. Similar to the greedy decoding case, our method achieves a ~1.5× speedup without compromising performance and a ~1.8× speedup with a modest 1-point BLEU/ROUGE score reduction with sampling.

| Dataset | IWSLT | | XSUM | |
|---|---|---|---|---|
| | BLEU | Speedup | ROUGE-L | Speedup |
| Vanilla Inference | 39.24 | - | 34.00 | - |
| BiLD | 39.72 (+0.48) | 1.51× | 34.34 (+0.34) | 1.22× |
| | 39.26 (+0.02) | 1.63× | 34.04 (+0.04) | 1.45× |
| | 38.27 (-0.97) | 1.80× | 33.10 (-0.90) | 1.85× |

**Less fallbacks:** The dynamic window size enables the small model to persist in making predictions when it is confident, thus minimizing the unnecessary engagement of the large model. This is supported by Table 4 where BiLD involves fewer number of fallbacks (23.24% → 21.09% and 36.84% → 32.33%) than [38, 4] while achieving better performance.

**Less rollbacks/rejections:** The dynamic window size further enables preemption of the small model when it is uncertain, which avoids rollback of the small model's wrong predictions. This is also supported by Table 4 where BiLD involves significantly fewer number of rollbacks (9.81% → 1.56% and 24.24% → 6.41%) than [38, 4] while achieving better performance.

Minimizing both fallbacks and rollbacks/rejections reduces unnecessary computation which directly translates to end-to-end latency improvement.

### 7.4 BiLD with Sampling

Our approach isn't restricted to greedy decoding, but it can seamlessly extend to sampling methods. The only modification is to perform random sampling instead of greedy sampling when drawing a token from both the small model and the large model while using the same fallback and rollback policy. This is because both the fallback and rollback policies, based on the maximum prediction probability, serve as an effective indicator of the small model's uncertainty in prediction and potential inaccuracies, regardless of the sampling method. The following table illustrates the latency versus performance trade-off of the sampling-based approach, specifically using nucleus sampling with p=0.8, similar to [6]. This evaluation follows the same environment as other experiments outlined in the paper, and both cases involve aligned small models.

Table 5 exhibits the BLEU/ROUGE-L score of BiLD on the IWSLT and XSUM benchmarks as well as their relative speedup. As can be seen in the table, our method exhibits a similar trend to the greedy decoding case. It achieves a ~1.5× speedup without compromising performance and a ~1.8× speedup with a modest 1-point BLEU/ROUGE score reduction.

### 7.5 Additional Analysis

#### 7.5.1 Model Analysis of BiLD: FLOPs, MOPs, and Arithmetic Intensity

Figure 8 compares average FLOPs, MOPs (memory operations), arithmetic intensity, and the latency speedup of the vanilla inference and BiLD on the CNN/DailyMail benchmarks. For BiLD, we use the model with roughly the same ROUGE-L score as the vanilla inference, and all the numbers are normalized by the numbers of the vanilla inference. The figure illustrates that BiLD exhibits slightly higher FLOPs compared to the vanilla inference. This is due to the fact that the autoregressive and non-autoregressive executions have the same amount of FLOPs, and BiLD involves additional overhead of running the small model alongside. However, in the case of MOPs, BiLD demonstrates a significant ~5× reduction of memory operations. This can be attributed to the capability of BiLD to process multiple tokens with a single weight load, thereby enhancing token-level parallelism and maximizing data reuse. In contrast, this is not the case in the vanilla inference where a single weight load can only process a single token. Consequently, BiLD achieves a significantly higher arithmetic intensity, which is approximately 5 times larger than the vanilla inference. Arithmetic intensity [72] measures the number of arithmetic operations that can be performed per memory operation. Given that memory operations can contribute more to the overall inference latency than arithmetic operations in many Transformer decoding scenarios [32], decreasing memory operations

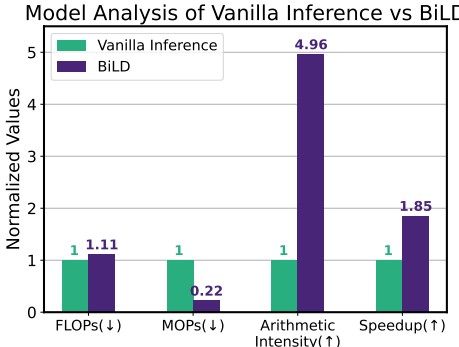

**Figure 8:** FLOPs, MOPs (memory operations), arithmetic intensity, and latency speedup comparison of vanilla inference and BiLD on the CNN/DailyMail benchmark. BiLD approach results in a remarkable reduction in MOPs due to the improved token-level parallelism, resulting in significantly higher arithmetic intensity.

| | |
|---|---|
| **Ground Truth** | And Siftables are an example of a new ecosystem of tools for manipulating digital information. |
| **Large** | And the Siftables are an example of a new generation of manipulation tools for digital data. |
| **Small** | And the if you look at the ifleses are an example of a new generation of technologies for manipulation of digital data. |
| **BiLD (ours)** | And the **Siftables** are an example of a new generation of **manipulation of** digital data. |

| | |
|---|---|
| **Ground Truth** | Which is great, because the Romans did not actually think that a genius was a particularly clever individual. |
| **Large** | That's great. The Romans didn't really think that a genius was a particularly smart individual. |
| **Small** | That's great. The tube didn't really think that a genius was a particularly lonely individual. |
| **BiLD (ours)** | That's great. The **Romans** didn't really think that **a** genius was a particularly smart individual. |

| | |
|---|---|
| **Ground Truth** | The viral particles then were released from the cells and came back and killed the E. coli. |
| **Large** | The viral particles then were released by the cells and came back and killed E. coli. |
| **Small** | The viral particles were then released by the cells and came back and killed E. Coke. |
| **BiLD (ours)** | The viral particles **then were** released **by the** cells and came back and killed E. **coli**. |

**Figure 9:** Example text sequences that BiLD generates with the validation set of IWSLT 2017 De-En, compared to the ground truths and the outputs of the large and small baselines. For BiLD, tokens generated by the large model are highlighted in red, while all the other tokens are generated by the small model. This illustrates that with a small engagement of the large model, BiLD can correct not only inaccurate vocabulary but also wrong semantics of the text that the small model would have otherwise generated.

and increasing arithmetic intensity can effectively alleviate the inference bottleneck. This leads to an overall latency speedup of $1.85\times$ on actual hardware.

### 7.5.2   Examples of Generated Sequences

Figure 9 provides examples of text sequences generated by BiLD on the validation set of IWSLT 2017 De-En, along with the ground truths (i.e., labels) and outputs of the pure large and small baseline models. The tokens generated from the large model of BiLD are highlighted in green, while all the other tokens are generated by the small model. The results illustrate that the small model often produces low-quality texts, by predicting inaccurate tokens which can alter the meaning of the entire sentence. To contrast, it is observed from the examples that BiLD is able to improve the text generation quality by letting the large model interrupt when the small model generates incorrect tokens. Particularly, in the examples provided, BiLD tends to be as strong as the large model at predicting terminologies. Overall, the large model's engagement in BiLD decoding not only improves the prediction accuracy but also prevents incorrect predictions from impacting the future ones.

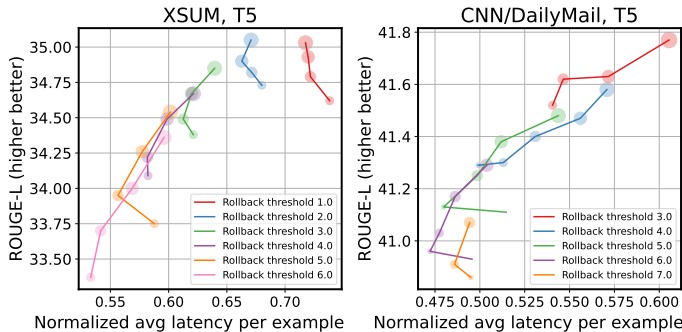

**Figure 10:** The trade-off between latency and generation quality (ROUGE-L) for the aligned BiLD model on two summarization tasks: (Left) XSUM and (Right) CNN/DailyMail. Each curve represents a different rollback threshold, with smaller thresholds indicating more rollbacks. The trade-off can be further obtained within each curve with different fallback thresholds, where larger scatter sizes indicate larger fallback thresholds. A larger fallback threshold implies more fallbacks.

### 7.5.3   Impact of Fallback and Rollback on Performance

We have explored how the BiLD framework can achieve different trade-offs between latency and generation quality by adjusting fallback and rollback thresholds. In this section, we present a detailed analysis of how these thresholds affect the performance using the aligned BiLD model on two different summarization tasks, XSUM and CNN/DailyMail, as illustrated in Figure 10. Different curves in the plot represent different rollback thresholds, and each scatter point within the curve represents different fallback thresholds. Note that a small rollback threshold implies more rollback, while a larger fallback threshold implies more fallback.

We observe a general trend where smaller rollback thresholds (i.e., more rollbacks) result in better generation quality but longer latency. This trend is expected because, with more rollback, we preempt more small model's predictions that can be potentially inaccurate by sacrificing the latency. Similarly, there is also a general trend that smaller fallback thresholds (i.e., fewer fallbacks) result in faster latency but a worse generation quality. However, we observed that lowering the fallback rates beyond a certain point can actually hurt both the latency and generation quality. This is because inaccurate predictions that the small model should have fallen back are later rolled back, incurring an extra 'flush' cost for the tokens that follow.

