# OpenReview forum: "Speculative Decoding with Big Little Decoder"
_NeurIPS.cc/2023/Conference — NeurIPS 2023 poster_

### Official Review · Reviewer_DxpA · 2023-06-11

**Soundness:** 4 excellent
**Presentation:** 4 excellent
**Contribution:** 4 excellent
**Rating:** 8
**Confidence:** 4

**Summary:**

This paper presents a new adaptive computation method to be used for text generation tasks. The method works by applying a small model, and if needed, revising its predictions using a larger, more expensive model. The resulting model obtains a far better speed-accuracy curve compared to the baseline model.

To put this work in a bit of context, this paper follows a rich line of work on adaptive computation in NLP. Most of these works explored the idea of early exiting. Many of them applied this approach to classification tasks (see [1-5] below for a few examples), but some also to generation tasks (e.g., [6] cited in this work, but also [7]). A much smaller number of works used a different implementation of this idea, by considering several models of different sizes. To the best of my knowledge, this has only been applied to classification tasks (e.g., [8-9]). A very recent work also compared these two approaches [10], also in a classification setup.

This work is the first work I’ve seen that applies the idea of combining multiple-models to generation tasks, and in this sense it is novel and interesting. The authors build on the standard adaptive computation framework (using confidence threshold to decide whether to use the larger model or not), and apply several other tricks to improve it: (a) using the large model in a non-autoregressive manner, to allow parallelization; (b) allowing the model to rollback “bad” generations; and (c) aligning the predictions of both models for better consistency between the two models. These ideas are interesting and a non-trivial contribution, but also hint that this approach requires some engineering to work (unlike the classification solutions, which generally work without much effort), and raise the question of how generalizable the proposed solution is. All-in-all, this is a strong paper that should be accepted to NeruIPS.

[1] DeeBERT: Dynamic Early Exiting for Accelerating BERT Inference - ACL Anthology
[2] [2004.02178] FastBERT: a Self-distilling BERT with Adaptive Inference Time
[3] [2004.07453] The Right Tool for the Job: Matching Model and Instance Complexities
[4] BERxiT: Early Exiting for BERT with Better Fine-Tuning and Extension to Regression - ACL Anthology
[5] [2104.08803] Consistent Accelerated Inference via Confident Adaptive Transformers
[6] [2207.07061] Confident Adaptive Language Modeling
[7] [1910.10073] Depth-Adaptive Transformer
[8] [2204.06271] TangoBERT: Reducing Inference Cost by using Cascaded Architecture
[9] [2012.14682] CascadeBERT: Accelerating Inference of Pre-trained Language Models via Calibrated Complete Models Cascade
[10] https://arxiv.org/abs/2306.02307


**Strengths:**

- An interesting and important problem
- A natural extension of a rich line of work (see discussion above)
- Impressive solutions to the challenges raised by this setup (see a-c above)
- Impressive results
- Interesting analysis and ablation experiments, hinting that all components contribute to the success of the approach.
- The paper is overall very clearly written, though see some comments below.



**Weaknesses:**

- It seems much engineering is required to make this method work as well. This is not such a strong critique, as the authors show it works for four different tasks, but still somewhat of a limitation.
- I am not entirely sure how the authors convert the decoder model to a non-autoregressive model. Do they make multiple decoding operations simultaneously? Or perhaps replace the decoder with an encoder? This feels like an important part of the construction, and requires some discussion and explanation. How much does it contribute to the performance gain?


**Questions:**

- Fig2: (a) what is the baseline (large) model performance? (b) why are we seeing a drop around 10-20%?
- Did the authors consider experiments with models from different families? Would this make the alignment problem harder?
- What is the performance of the small model in figure 4? This seems like an important baseline

---

> ### Author Rebuttal · Authors · 2023-08-10
>
> ### W1. Elaboration on non-autoregressive use of the model
>
> We can use the same decoder model, but rather in a non-autoregressive (parallel) fashion by providing the model with multiple tokens and processing them simultaneously.
> This mechanism is identical to how decoder models can simultaneously handle multiple tokens in a prompt, without necessitating architectural modifications.
> By using an upper diagonal attention mask, tokens are restricted to attending only to preceding tokens; therefore, running multiple decoding operations simultaneously maintains the same functionality as the autoregressive counterparts when processing the same tokens (hence resulting in no performance degradation) but only with a improved latency.
>
>
> ###  Q1. Performance of the small model
>
> Below, we provide a comparison of the small model to the larger counterpart and BiLD.
> We will elaborate on this point in the revised version of our paper.
>
>
> | Task   |                    | BLEU  | Latency |
> | ------ | ------------------ | ----- | ------- |
> | IWSLT  | Large              | 40.33 | 1x      |
> |        | Small              | 37.47 | 2.61x   |
> |        | BiLD (match BLEU)  | 40.33 | 1.43x   |
> |        | BiLD (BLEU - 1)    | 39.44 | 1.58x   |
> |        |                    |       |         |
> | WMT    | Large              | 31.38 | 1x      |
> |        | Small              | 28.74 | 2.50x   |
> |        | BiLD (match BLEU)  | 31.28 | 1.34x   |
> |        | BiLD (BLEU - 1)    | 30.47 | 1.43x   |
> |        |                    |       |         |
> | XSum   | Large              | 35.10 | 1x      |
> |        | Small              | 31.23 | 4.49x   |
> |        | BiLD (match ROUGE) | 35.12 | 1.48x   |
> |        | BiLD (ROUGE- 1)    | 34.02 | 1.72x   |
> |        |                    |       |         |
> | CNN/DM | Large              | 41.54 | 1x      |
> |        | Small              | 40.43 | 4.15x   |
> |        | BiLD (match ROUGE) | 41.44 | 1.71x   |
> |        | BiLD (ROUGE- 1)    | 40.57 | 2.05x   |
>
>
> ### Q2. Elaboration on Figure 2
> The large model performance is the right-most scatter (100% Large Model) on each plot, which is around 31.3 and 41.5 for WMT and CNN/DM, respectively.
> The drop around 10-20% is due to the increased engagement of the small model in text generation, which converges to the small model's own performance on the left-most scatter (0% Large Model).
>
>
> ### Q3. Experiments on different families of models
>
> Although we have yet to experiment with models beyond two model families, T5 and mT5, in our experiments, we believe that extending the proposed framework to different small and large model combinations is a straightforward translation given the strong generalizability across different models and tasks we presented in our paper. Investigating generalizability across varied models and scales, along with exploring the effects on model alignment, is indeed an intriguing avenue for future research.

---

> > ### Comment · Reviewer_DxpA · 2023-08-19
> >
> > Thank you for the clarifications, I am happy with the responses.

---

### Official Review · Reviewer_asbG · 2023-06-27

**Soundness:** 3 good
**Presentation:** 2 fair
**Contribution:** 2 fair
**Rating:** 4
**Confidence:** 4

**Summary:**

The paper proposes BiLD, a framework to coordinate a large model and a small model such that the large model is only executed infrequently and efficiently in a non-autoregressive manner to refine the small model's inaccurate predictions. The proposed two policies allows the small model to hand control over to the large model if it is not confident enough and the large model to review and correct the small model's inaccurate predictions.



**Strengths:**

- The proposed fallback method is novel in multiple ways: 1) use small model's generation as part of big model's input, thus transforming an autoregressive generation problem to semi-autoregressive. 2) Improves compute intensity as tokens can be generated in parallel in the big model, which could significantly reduce latency while not sacrificing model quality much.

- Rollback policy mitigate the issue of small model being over confident.

**Weaknesses:**

- Evaluation benchmarks are mostly on translation, not on QA tasks. The paper should evaluate more QA tasks including Triviaqa, Squad, webqs and Nqs.

- The paper should compare with CALM [1] to demonstrate the effectiveness of big/small model with fallback and rollback rather than having early stopping in a per layer basis.

- Inference batch size of 1 is quite limited. Will this approach work for batch size larger than one? If the batch size is larger than one, there will be a parameter loading overhead and very likely both models need to be loaded. Loading parameters can dominate the latency. Batch size of one makes serving easier for any conditional computation approaches, however, it can lead to under utilization of hardware. Could you provide some evidence of batch size larger than one?

[1]: https://arxiv.org/abs/2207.07061

**Questions:**

1. How to set the thresholding to fallback? The paper briefly mentions that the small model can be over confident therefore, even with a high confidence, the prediction can be wrong.

2. How large is the big model compared to the small model? Any restrictions?

**Limitations:**

Not addressed.

---

> ### Author Rebuttal · Authors · 2023-08-10
>
> ### W1. Evaluation on QA Benchmarks
>
> We acknowledge the comment that our paper lacks evaluations on QA tasks.
> However, we highlight that **QnA tasks are not the ideal benchmarks for assessing our methodology's effectiveness**. This is because the main focus of our framework lies in enhancing *decoding processes* to efficiently generate lengthy sequences. In contrast, QnA benchmarks, often involving multiple-choice or short-answer questions (Triviaqa, Squad, Webqs and Nqs), mostly entail brief decoding lengths.
> Therefore, these benchmarks primarily evaluate the ability of a given language model to encode and understand the question prompt, rather than its ability to generate a response, making them unsuitable for evaluating decoding optimization methodologies like BiLD.
>
>
> ### W2. Comparison to CALM
>
> Figure 7 (Appendix 6.2.2) contains the information this comment is requesting for. We directly compared CALM applied to the mT5-small model with BiLD applied to the same model. The first layer of the pre-trained BiLD model serves as our small model, but this choice is one of many design points and other choices are viable as well. Choosing the first layer of the large model as the small model showcases the framework's extensibility when only a single model is accessible while preserving the core idea of using both big and small models.
>
> ### W3. Multi-batch Inference
>
> While we only evaluate the use case of batch size 1, it is **not a limitation and it is straightforward in modern LLM serving systems to extend the framework to multi-batch applications**. This is because the modern LLM serving systems (especially for multi-batch applications) such as HuggingFace TGI are equipped with “iteration-level batching” [1, 2]. The main idea is to improve the throughput of LLM inference by removing “bubbles” when batching multiple examples with different lengths, and this is critical in many applications as different requests in a batch could have different numbers of prompt tokens and decoding iterations, resulting in some requests finishing earlier than the others. Iteration-level batching prevents this by storing the decoding jobs in a request pool in the iteration granularity and dynamically batching multiple examples on different decoding iterations together in every decoding iteration.
>
> Therefore, BiLD (as well as other speculative decoding frameworks) can be supported seamlessly on top of the existing inference system, with the only difference of having two separate request pools one for the large model, and the other for the small model. For example, even though one example A falls back at its 2nd step and the other B falls back at its 3rd step due to different low-confidence positions, the query will be handled as follows.
>
> - 1st and 2nd iteration: A and B are batched together from the small model’s request pool
> - 3rd iteration:
>     - A will be registered to the large model’s request pool, from which it will be batched with other jobs in the same request pool for its 3rd decoding iteration.
>     - B is still at the small model’s request pool and can be batched together with other jobs at the same request pool if there are any.
>     - Note that within the iteration-level batching system, neither A nor B needs to be at the same decoding step with the other examples that are batched together
>
>
>
> ### Q1. How to set the thresholding to fallback and Over-confidence of the small model
>
> The fallback threshold is explored within the hyperparameter space outlined in Appendix 7.1.2. It is possible that the small model to be overconfident and the fallback threshold might not always detect such occurrences; however, this doesn't impact text generation quality as such occurrences are fixed by the large model during the rollback procedure.
>
> ### Q2. How large is the big model compared to the small model?
>
> This is indeed a very interesting question, and finding the right-scaled small model given a large model with the optimal performance tradeoff can be an open-ended research direction.
> This is indeed a very interesting question, and identifying the appropriately scaled small model to a given large model for the optimal performance tradeoff can lead to an open-ended avenue of research.
>
> Below, we present a comparison table that combines (non-aligned) mT5-base and mT5-small to mT5-large. While the base model enhances the lower threshold of attainable text generation performance, the considerable speedup it offers offsets this gain, leading to a less favorable performance-latency trade-off. Across our experiments, we recognize the significance of maintaining the small model's latency at a sufficiently low level to achieve a favorable balance. We have observed that setting the small model size to approximately 10 times smaller than the large model size serves as a reasonable guideline in this regard. However, delving into a systematic analysis of scaling rules and developing decision-making policies would be an interesting future research direction.
>
>
> |                      | BLEU  | Latency |
> | -------------------- | ----- | ------- |
> | Large                | 40.32 | 1x      |
> | Base                 | 38.13 | 1.9x    |
> | Small                | 36.61 | 2.7x    |
> |                      |       |         |
> | Base + Large         | 40.30 | 1.19x   |
> | **Small + Large (BiLD)** | **40.33** | **1.43x**   |
>
>
>
> [1] Yu et al. Orca: Progressive Learning from Complex Explanation Traces of GPT-4
>
> [2] Kwon et al. vLLM: Easy, Fast, and Cheap LLM Serving with PagedAttention

---

> > ### Comment · Reviewer_asbG · 2023-08-19
> > **Acknowledge the reponse**
> >
> > Thanks for the rebuttal. The reviewer still feels CALM and method in this paper are fundamentally similar. Token-level routing is not practical for serving when batch size is greater than 1 while sequence level routing is not illustrated effective yet. The reviewer does not agree that QA benchmarks are not relevant to this method, as QA benchmarks are more challenging to get good scores when model capacity is limited.

---

> > > ### Author Response · Authors · 2023-08-21
> > > **Additional comments on token-level batching and comparison with CALM**
> > >
> > > We appreciate the reviewer for the further comments. We would like to make further clarification to the points brought up in the comments.
> > >
> > > ### 1. Token-level Batching/Routing
> > >
> > > > Token-level routing is not practical for serving when batch size is greater than 1 while sequence level routing is not illustrated effective yet.
> > >
> > > We politely claim that token-level batching/routing (or iteration-level batching) is **a more efficient and pragmatic solution** compared to basic batching. This method is **already a standard in various LLM serving systems**, including open-source frameworks like Huggingface’s Text Generation Inference (TGI) [1], vLLM [2], Periflow [3], and FlexFlow [4], as well as in proprietary systems.
> > >
> > > Additionally, as highlighted by Reviewer mW2r, **integrating the BiLD framework into token-level batching systems is a straightforward extension**, primarily requiring engineering work rather than fundamental changes. For instance, FlexFlow, a recent framework, supports speculative decoding, another way of coordinating multiple-sized models in run-time, on top of their token-level batching system. This emphasizes that the BiLD framework can be **seamlessly supported on modern batching systems**.
> > >
> > > [1] TGI, github: huggingface/text-generation-inference
> > >
> > > [2] vLLM, github: vllm-project/vllm
> > >
> > > [3] Periflow, github: friendliai/periflow-client
> > >
> > > [4] FlexFlow, github: flexflow/FlexFlow
> > >
> > >
> > > ### 2. Comparison with CALM
> > >
> > > > The reviewer still feels CALM and method in this paper are fundamentally similar.
> > >
> > > We've incorporated a requested comparison table between CALM and BiLD. Note that since BiLD consists of a small and a large model, we have utilized the *large model* for CALM to ensure a fair comparison. We fine-tuned the mT5-large model on IWSLT using CALM loss objective (weighted per-layer loss average) with the same hyperparameters in the paper. Furthermore, we have conducted an exhaustive hyperparameter search to determine the optimal inference setup, following the methodology outlined in our paper.
> > >
> > >
> > >
> > > | Method               | BLEU  | Norm. Latency | CALM avg. layer |
> > > | -------------------- | ----- | ------------------ | ---------------------------- |
> > > | Baseline (mT5-large) | 40.32 | 1x                 |                              |
> > > |                      |       |                    |                              |
> > > | BiLD                 | **40.54** | **1.23x**              |                              |
> > > |                      | **39.87** | **1.49x**              |                              |
> > > |                      |       |                    |                              |
> > > | CALM                 | 35.03 | 1.05x              | 8.9                          |
> > > |                      | 32.67 | 1.18x              | 5.2                          |
> > > |                      | 31.33 | 1.25x              | 3.6                          |
> > >
> > >
> > > The table clearly shows **BiLD's substantial superiority over CALM**. We hypothesize two reasons for CALM's reduced performance using the 24-layer large model (note that the original CALM paper only assessed an 8-layer small model):
> > >
> > >
> > > 1. Unlike in shallower models, deeper models' early layers might lack the representative power of the full model, making it more challenging to match predictive performance within a few initial layers. This is clear in the table, where utilizing only 3-5 layers leads to a notable BLEU drop of 8-9 points.
> > >
> > > 2. CALM involves early exiting decisions in every layer, introducing notable latency overhead due to frequent dense multiplication with a huge vocabulary embedding. This overhead is particularly exaggerated when the vocabulary size is big. For instance, the table indicates that running ~9 layers yields similar latency as the full model, demonstrating a huge overhead regarding the early exiting policy. In contrast, BiLD's fallback policy infrequently runs at per-token granularity and can be more latency efficient.

---

### Official Review · Reviewer_wHu9 · 2023-07-05

**Soundness:** 3 good
**Presentation:** 3 good
**Contribution:** 3 good
**Rating:** 6
**Confidence:** 4

**Summary:**

The paper presents an approach to improve decoding using a smaller model to create drafts, and fallback to a larger model whenever the smaller model predicts any token with probability smaller than a fixed threshold. Additionally, during this fallback, if the larger model produces a token that is not in sync with the prediction from the smaller model, the predictions are rolled back to the point until the predictions are in sync and then prediction continues using the predicted token from the larger model.

Given that the larger model can use causal attention masking and predict tokens in parallel using the predictions from the smaller model as hints for teacher forcing, the whole process becomes latency efficiency (at the cost of flops).

The method is inspired from `Accelerating Large Language Model Decoding with Speculative Sampling`, which too uses two models. The small one for creating draft predictions using non-autoregressive decoding, and the second (larger) model for iterating over the predictions from the smaller model using the larger model.

**Strengths:**

The method is an improvement over speculative sampling in terms of latency by not just using the small model for the initial prediction, but using a procedure that keeps shifting control back and forth between the smaller and the larger model.

Experiments show that this significantly improves the speedup compared to speculative sampling.

**Weaknesses:**

It is unclear how this can achieve quality better than speculative sampling.

Speculative sampling just uses the small model as draft. The draft is refined by the larger model, but the predictions are always going to be the same as that of the large model.

However, that is not the case with this paper. In this paper, all the predictions may come from the small model if the model never produces a token with low probability.

If that is the case, I would expect the methodology in this paper to only outperform Speculative sampling in terms of latency. Quality should be at par or below the larger model, but seems like that is not the case.

**Questions:**

As mentioned in the weakness section, what are your thoughts on why the model can perform superior compared to Speculative sampling?

**Limitations:**

No limitations that I can think of, and none discussed in the paper.

---

> ### Author Rebuttal · Authors · 2023-08-10
>
>
> ### W1. Insights on Better Latency and Performance
>
> Quantitative analysis reveals a consistent trend where our method, when compared to speculative decoding [1, 2], effectively enhances both text generation quality and latency. We provide insights and explanations into why our approach surpasses speculative decoding frameworks.
>
>
> - **(1) Better text generation quality:**
>     - (1-1) **Ensembling effect:** The power of blending outputs from multiple models has been well-explored in various fields. This is also the case in open-source LLM models which exhibit diverse strengths and weaknesses due to variations in data, architectures, and hyperparameters, making different models complementary to each other [Ref: LLM-Blender]. In fact, we show such effects of blending multiple model outputs in Figure 2, where a combination of 20% of the large model's prediction with the small model's prediction outperforms the exact imitation of the large model's behavior. Our approach offers fallback and rollback policies that efficiently exploit optimal ensemble point, which produces superior output quality compared to the unbiased estimate of the large model in [1, 2].
>     - (1-2) **Rejection policy that leads to higher predictive performance:** Our method adopts a rejection policy that *completely* discards the small model's prediction if it significantly deviates from the large model's counterpart, based on cross entropy-based distance metric. This contrasts with speculative decoding, where the decision involves a stocastic rejection sampling process. We empirically observe that BiLD’s “hard rejection policy” allows better BLEU/ROUGE score with significantly fewer number of rejections (rollbacks) than the “stocastic rejection policy” of speculative decoding as described in the table below. We hypothesize that this boost in predictive performance stems from our hard rollback policy, which prevents potentially erroneous predictions by ruling out stochasticity. We additionally hypothesize that such a strategy can addresses exposure bias, mitigating the impact of a single early-stage misprediction on subsequent predictions.
> - **(2) Lower end-to-end latency:**
>     - Furthermore, our fallback policy introduces a dynamic window size (i.e. number of small model's consecutive iterations) that is determined based on the run-time prediction confidence of the small model. This is in contrast with speculative decoding which adopts a static window size. The advantages of the dynamic window size is in two-fold:
>         - (2-1) **Less fallback:** The dynamic window size enables the small model to persist in making predictions when it is condifent, thus minimizing the unnecessary engagement of the large model. This is supported by the table below where BiLD involves fewer number of fallbacks (23.24 → 21.09 and 36.84 → 32.33) than speculative sampling while achieving better performance.
>         - (2-2) **Less rollback/rejection:** The dynamic window size further enables preemption of the small model when it is uncertain, which avoids rollback of the small model’s wrong predictions. This is also supported by the table below where BiLD involves significantly fewer number of rollbacks (9.81 → 1.56 and 24.24 → 6.41) than speculative sampling while achieving better performance.
>
>         Minimizing both fallback and rollback/rejection reduces unnecessary computation which directly translates to end-to-end latency improvement.
>
>
> | Task  | Method              | BLEU/ROUGE | Latency | % Fallback | % Rollback (rejection) |
> | ----- | ------------------- | ---------- | ------- | ---------- | ---------------------- |
> | IWSLT | Speculative         | 39.93      | 1.28x   | 23.24%     | 9.81%                  |
> |       | BiLD (Better BLEU)  | **40.33**  | 1.43x   | 21.09%     | **1.56%**              |
> |       |                     |            |         |            |                        |
> | XSum  | Speculative         | 35.00      | 1.25x   | 36.84%     | 24.24%                 |
> |       | BiLD (Better ROUGE) | **35.12**  | 1.48x   | 32.33%     | **6.41%**              |
>
>
> ### W2. Can all the predictions come from the small model?
>
> Although it is highly unlikely for the small model to generate all tokens with high probability, we prevent this occurrence by limiting the maximum consecutive predictions from the small model to 10 tokens. This is also stated in the Appendix for reference.
>
>
> [1] Chen et al. Accelerating Large Language Model Decoding with Speculative Sampling
>
> [2] Leviathan et al. Fast Inference from Transformers via Speculative Decoding

---

### Official Review · Reviewer_mW2r · 2023-07-08

**Soundness:** 3 good
**Presentation:** 3 good
**Contribution:** 2 fair
**Rating:** 6
**Confidence:** 4

**Summary:**

This paper presents an efficient generation approach that utilizes a smaller model during decoding to reduce computational requirements. In essence, the large model is only employed when the small model's generation exhibits low confidence (fallback policy) or when there is a significant distribution difference between the previously generated tokens of the small and large models (rollback policy). The authors demonstrate that this approach achieves a speedup of up to 2X for downstream tasks using FLAN-T5 models. However, a potential limitation of this approach is its scalability issue when the batch size exceeds 1, which may impede its practical applicability. Additionally, there is a lack of discussion regarding previous works proposing similar ideas, which would provide valuable context and comparisons.

**Strengths:**

- The paper is well written, presenting a clear and easily understandable explanation of the proposed method, which is both straightforward and effective. As generation inference continues to gain importance, this paper offers a valuable solution to enhance performance in that regard.
- The paper effectively illustrates the tradeoff between efficiency and performance through the figures presented. The visual representations provide a clear understanding of how the proposed method strikes a balance between the two aspects.

**Weaknesses:**

- **Batch size:** The biggest limitation of the approach is that it probably only works for a batch size of 1 (the authors only tested this setting as well). Extending the approach to handle multiple examples within a batch poses challenges due to the significant variation in low-confidence positions across the generations for different examples. Consequently, it becomes difficult to envision the practical usability of the approach, particularly considering the importance of batching for efficient hardware utilization. It is crucial for the authors to conduct experiments to ascertain whether the method retains its advantages when the batch size exceeds 1 and provide a more comprehensive discussion on this aspect.
- **Novelty and comparison to previous work:** it is important to note that the core idea of this work bears significant resemblance to existing work [1]. Therefore, a quantitative comparison and discussion concerning the relationship between this work and the aforementioned existing work is necessary.

[1] Fast Inference from Transformers via Speculative Decoding, Leviathan et al., 2023

**Questions:**

- How does the method generalize to models of different scales? Given a large model, how to most effectively select a corresponding small model? A discussion of scales would be very helpful.
- The main results presented in Table 1 and Figure 4 lack benchmarking for the small model's performance and speedup. It would be helpful to compare the performance of the small model against its larger counterpart to provide a baseline for understanding the achieved improvements.
- The evaluation of the study focuses on in-domain tasks. Would the method provide advantages for out-of-domain generalizations? For examples, tasks that are not trained with FLAN, e.g., GSM8K?
- The fallback policy involves regeneration for (n-m) tokens and appears to be computationally expensive. It would be insightful to provide a more detailed analysis, such as the percentage of occurrences of rollback and fallback events. Similarly, an analysis indicating the percentage of predictions made by the small model versus the large model would be insightful as well.

**Limitations:**

The authors did not discuss the limitations of the work explicitly. However, I believe that the work appears to have minimal negative societal impact.

---

> ### Author Rebuttal · Authors · 2023-08-10
>
> ### W1. Batch sizes
>
> While we only evaluate the use case of batch size 1, it is **not a limitation and it is straightforward in modern LLM serving systems to extend the BiLD framework to multi-batch applications**. This is because the modern LLM serving systems (especially for multi-batch applications) such as HuggingFace TGI are equipped with “iteration-level batching” [1, 2]. The main idea is to improve the throughput of LLM inference by removing “bubbles” when batching multiple examples with different lengths, and this is critical in many applications as different requests in a batch could have different numbers of prompt tokens and decoding iterations, resulting in some requests finishing earlier than the others. Iteration-level batching prevents this by storing the decoding jobs in a request pool in the iteration granularity and dynamically batching multiple examples on different decoding iterations together in every decoding iteration.
>
> Therefore, BiLD (as well as other speculative decoding frameworks) can be supported seamlessly on top of the existing inference system, with the only difference of having two separate request pools one for the large model, and the other for the small model. For example, even though one example A falls back at its 2nd step and the other B falls back at its 3rd step due to different low-confidence positions, the query will be handled as follows.
>
> - 1st and 2nd iteration: A and B are batched together from the small model’s request pool
> - 3rd iteration:
>     - A will be registered to the large model’s request pool, from which it will be batched with other jobs in the same request pool for its 3rd decoding iteration.
>     - B is still at the small model’s request pool and can be batched together with other jobs at the same request pool if there are any.
>     - Note that within the iteration-level batching system, neither A nor B needs to be at the same decoding step with the other examples that are batched together
>
> ### W2. Quantitative Comparison to Speculative Decoding
>
> In the table below, we present a comprehensive quantitative comparison between our method and speculative decoding [3, 4] across two different datasets: IWSLT for machine translation and XSum for summarization. We will include this comparison in the revised paper. In order to ensure a fair comparison that isolates the impact of the frameworks themselves, we employ the baseline (non-aligned) small model for both BiLD and speculative decoding. We maintained the same evaluation setup and hyperparameter space that are outlined in Appendix 6.1.
>
> The table below shows two BiLD configurations across two different tasks and two different sampling methods of greedy sampling and nucleus sampling with p=0.8, and temperature of T=1.0 as in [4]. In all cases, the first row of BiLD matches latency, and the second row matches BLEU/ROUGE-L scores as speculative decoding. Across all experiments, BiLD consistently outperforms speculative decoding. It achieves either **(1) notably improved BLEU/ROUGE scores of up to an additional 2 points with equivalent latency gains, or (2) superior latency gains of up to +0.4x while retaining the same score.**
> We will further elaborate on this point in the revised version.
>
> **IWSLT**
>
> | Sampling | Method  | BLEU  | Norm. Latency |
> | -- | -- | -- | -- |
> | Greedy   | Baseline  | 40.32 | 1x   |
> | | Speculative | 39.93 | 1.28x  |
> |  | BiLD (match latency) | 40.54 | 1.23x  |
> |  | BiLD (match BLEU) | 39.87 | 1.49x |
> | Sampling | Baseline | 39.24 | 1x |
> |  | Speculative | 38.79 | 1.36x |
> |  | BiLD (match latency) | 39.56 | 1.35x |
> |  | BiLD (match BLEU) | 38.88 | 1.49x |
>
> **XSum**
>
> | Sampling | Method  | ROUGE-L | Norm. Latency |
> | -- | -- | -- | -- |
> | Greedy | Baseline | 35.08 | 1x |
> |  | Speculative | 35.00 | 1.25x  |
> |  | BiLD (match latency) | 35.30  | 1.42x  |
> |  | BiLD (match ROGUE) | 34.96 | 1.50x  |
> | Sampling | Baseline | 34.02   | 1x  |
> |  | Speculative | 32.32 | 1.23x |
> |  | BiLD (match latency) | 34.32   | 1.22x |
> | | BiLD (match ROGUE) | 32.81 | 1.61x |
>
> ### Q1: How to most effectively select a small model
>
> We have provided a detailed discussion in [Q2 of Reviewer asbG](https://openreview.net/forum?id=EfMyf9MC3t&noteId=nrwJEv4oRT).
>
> ### Q2. Performance of the small model
>
> We have provided the small model's performance in [Q1 of Reviewer DxpA](https://openreview.net/forum?id=EfMyf9MC3t&noteId=8Hcp16yZeo).
>
> ### Q3. More Statistics
>
> We present an in-depth comparison of rollback and fallback event percentages in two different datasets, using the unaligned small model. Please note that % fallbacks indicates % large model.
>
> - With a fallback engagement ranging from 20% to 30% (i.e. 20-30% engagement of the large model), BiLD attains performance on par with the baseline, which resonates with our observations in Figure 2.
> - The occurrence of rollback in the BiLD framework is infrequent and is ~2% for IWSLT and ~6% for XSum. Consequently, the overhead incurred from regenerating (n-m) tokens remains small. While a similar regeneration mechanism exists in the speculative decoding framework [3, 4], our approach demonstrates notably lower rollback occurrences. In the speculative decoding, the % rollback/rejection is around 10% for IWSLT and 24% for XSum. Such a reduced rollback overhead in our framework directly translates to enhanced speedup.
>
> | Task  | Method  | BLEU/ROUGE | Latency | % Fallback | % Rollback |
> | -- | -- | -- | -- | -- | -- |
> | IWSLT | Speculative  | 39.93 | 1.28x | 23.24%  | 9.81%  |
> | | BiLD  | 40.33 | 1.43x | 21.09%  | 1.56% |
> | XSum  | Speculative | 35.00      | 1.25x | 36.84% | 24.24% |
> |  | BiLD  | 35.12  | 1.48x | 32.33% | 6.41% |
>
> [1] Yu et al. Orca: Progressive Learning from Complex Explanation Traces of GPT-4
>
> [2] Kwon et al. vLLM: Easy, Fast, and Cheap LLM Serving with PagedAttention
>
> [3] Leviathan et al. Fast Inference from Transformers via Speculative Decoding
>
> [4] Chen et al. Accelerating Large Language Model Decoding with Speculative Sampling

---

> > ### Comment · Reviewer_mW2r · 2023-08-18
> > **Thanks for the author response**
> >
> > I appreciate the author's thoughtful response, which has helped clarify several points.
> >
> > Regarding the first issue, I agree that batch size would be less problematic when utilizing iteration-level batching. However, the paper would be strengthened substantially if the authors could include quantitative speedup comparisons between BiLD and standard decoding with this implementation. Nonetheless, the batch size=1 results adequately demonstrate the approach's efficacy. Increasing batch sizes greater than 1 appears to require primarily engineering efforts rather than fundamental enhancements.
> >
> > For W2, thanks for providing the additional results. Given that, could you provide an intuitive understanding as to why BiLD provides additional efficiency gains compared to speculative decoding?
> >
> > Based on the authors thoroughly addressing the majority of the questions and concerns raised in my review, I have made the decision to increase my score to 6.

---

> > > ### Author Response · Authors · 2023-08-19
> > > **Further Comments**
> > >
> > > Thank you for your valuable comments and feedback.
> > >
> > > Regarding the question of why BiLD outperforms speculative decoding, we offer insightful explanations in our response to [reviewer wHu9](https://openreview.net/forum?id=EfMyf9MC3t&noteId=NJBmWF2n8y).
> > > This can be summarized as **(1) better text generation quality** due to the ensembling effect and BiLD's rejection policy that leads to higher predictive performance as well as (2) **lower end-to-end latency** due to less fallback and rollback/rejections.
> > >
> > > Below is a more detailed discussion on each point.
> > >
> > > (1) **Better text generation quality:**
> > >
> > > (1-1) **Ensembling effect:** The power of blending outputs from multiple models has been well-explored in various fields. This is also the case in open-source LLM models which exhibit diverse strengths and weaknesses due to variations in data, architectures, and hyperparameters, making different models complementary to each other [3]. In fact, we show such effects of blending multiple model outputs in Figure 2, where a combination of 20% of the large model's prediction with the small model's prediction outperforms the exact imitation of the large model's behavior. Our approach offers fallback and rollback policies that efficiently exploit optimal ensemble point, which produces superior output quality compared to the unbiased estimate of the large model in [1, 2].
> > >
> > > (1-2) **Rejection policy that leads to higher predictive performance:** Our method adopts a rejection policy that completely discards the small model's prediction if it significantly deviates from the large model's counterpart, based on cross entropy-based distance metric. This contrasts with speculative decoding, where the decision involves a stocastic rejection sampling process. We empirically observe that BiLD’s “hard rejection policy” allows better BLEU/ROUGE score with significantly fewer number of rejections (rollbacks) than the “stocastic rejection policy” of speculative decoding as described in the table below. We hypothesize that this boost in predictive performance stems from our hard rollback policy, which prevents potentially erroneous predictions by ruling out stochasticity. We additionally hypothesize that such a strategy can addresses exposure bias, mitigating the impact of a single early-stage misprediction on subsequent predictions.
> > >
> > > (2) **Lower end-to-end latency:**
> > > Furthermore, our fallback policy introduces a dynamic window size (i.e. number of small model's consecutive iterations) that is determined based on the run-time prediction confidence of the small model. This is in contrast with speculative decoding which adopts a static window size. The advantages of the dynamic window size is in two-fold:
> > >
> > > (2-1) **Less fallback:** The dynamic window size enables the small model to persist in making predictions when it is condifent, thus minimizing the unnecessary engagement of the large model. This is supported by the table below where BiLD involves fewer number of fallbacks (23.24 → 21.09 and 36.84 → 32.33) than speculative sampling while achieving better performance.
> > >
> > > (2-2) **Less rollback/rejection:** The dynamic window size further enables preemption of the small model when it is uncertain, which avoids rollback of the small model’s wrong predictions. This is also supported by the table below where BiLD involves significantly fewer number of rollbacks (9.81 → 1.56 and 24.24 → 6.41) than speculative sampling while achieving better performance.
> > > Minimizing both fallback and rollback/rejection reduces unnecessary computation which directly translates to end-to-end latency improvement.
> > >
> > >
> > > | Task  | Method              | BLEU/ROUGE | Latency | % Fallback | % Rollback (rejection) |
> > > | ----- | ------------------- | ---------- | ------- | ---------- | ---------------------- |
> > > | IWSLT | Speculative         | 39.93      | 1.28x   | 23.24%     | 9.81%                  |
> > > |       | BiLD (Better BLEU)  | **40.33**  | 1.43x   | 21.09%     | **1.56%**              |
> > > |       |                     |            |         |            |                        |
> > > | XSum  | Speculative         | 35.00      | 1.25x   | 36.84%     | 24.24%                 |
> > > |       | BiLD (Better ROUGE) | **35.12**  | 1.48x   | 32.33%     | **6.41%**              |
> > >
> > >
> > >
> > > Reference:
> > > [1] Chen et al. Accelerating Large Language Model Decoding with Speculative Sampling
> > >
> > > [2] Leviathan et al. Fast Inference from Transformers via Speculative Decoding
> > >
> > > [3] Jiang at el. LLM-Blender: Ensembling Large Language Models with Pairwise Ranking and Generative Fusion

---

### Official Review · Reviewer_NSPU · 2023-07-22

**Soundness:** 2 fair
**Presentation:** 3 good
**Contribution:** 2 fair
**Rating:** 5
**Confidence:** 4

**Summary:**

The paper proposes Big Little Decoder (BiLD), a framework to accelerate autoregressive text generation by using a small decoder model to generate most tokens efficiently and invoking a large decoder model occasionally to refine inaccurate predictions. BiLD manually designs two policies to invoke refinement, 1) a fallback policy (based on the small model's uncertainty) to hand control from the small to large model when uncertain, and 2) a rollback policy (based on the distance of two models' probability) for the large model to correct inaccurate small model predictions. Experiments on machine translation and summarization tasks show BiLD can achieve up to 2x speedup on a GPU with minimal quality loss.

**Strengths:**

1. The idea of combining small and large models is intuitive and aligns with the observation that small models can achieve comparable quality if a few inaccurate predictions are corrected.

2. The fallback and rollback policies are simple yet effective mechanisms to coordinate the models and empirically show better performance than speculative sampling.

3. The framework is general and does not require additional training, while the prediction alignment technique further improves BiLD's performance with minimal extra effort.

**Weaknesses:**

1. A few very similar prior works are not cited and discussed [2-3], i.e., the idea of using a fast-inference model to generate tokens quickly and refining with a large model.

2. The paper criticizes [1] as "suggests an unbiased estimator that recovers the exact probability distributions of the more powerful model, it can still have a large variance that may impact current and future predictions". But the proposed BiLD approach itself even does not have any guarantee of unbiased or low-biased recovery of the large model's probability. It seems that BiLD can only be used for greedy decoding, rather than arbitrary sampling from the large model's distribution, which is quite important in a few downstream tasks [6].

3.  The policies require tuning fallback and rollback thresholds, adding hyperparameters. Optimal values likely vary across tasks and models.

4. While sharing several common techniques to the non-autoregressive transformer (NAT) models, the literature on NAT is not discussed. For example, the prediction alignment technique is very similar to knowledge distillation adopted to train NAT models [4].

5. The proposed method is only evaluated on the machine translation and text summarization datasets, which have been shown to not that hard for small or non-autoregressive sequence models [4-5]. More challenging datasets, such as HumanEval used in [1], should be evaluated.

6. A minor problem is that there are limited insights into what types of predictions the large model tends to correct.

[1] Accelerating Large Language Model Decoding with Speculative Sampling

[2] Lossless Acceleration for Seq2seq Generation with Aggressive Decoding

[3] Blockwise Parallel Decoding for Deep Autoregressive Models

[4] Levenshtein Transformer

[5] Deep Encoder, Shallow Decoder: Reevaluating Non-autoregressive Machine Translation

[6] Self-Consistency Improves Chain of Thought Reasoning in Language Models

**Questions:**

Questions:

1. In Figure 2, why does 20% achieve better performance than 40 or 80 percent?

Typos:

line 142: regressive ==> autoregressive

---

> ### Author Rebuttal · Authors · 2023-08-10
>
> ### W1. BiLD with Random Sampling
>
> Our approach **isn't restricted to greedy decoding, but it seamlessly extends to sampling methods.** Even though the fallback policy is based on the maximum prediction probability of a small model, we can still perform decoding based on random sampling. The only modification is to perform random sampling instead of greedy sampling when drawing a token from both the small model and the large model. We keep the same fallback and rollback policy. This is because both the fallback and rollback policies, based on the maximum prediction probability, serve as an effective indicator of the small model's uncertainty in prediction and potential inaccuracies, regardless of the sampling method. The following table illustrates the latency versus performance trade-off of the sampling-based approach, specifically using nucleus sampling with p=0.8, similar to [1]. This evaluation follows the same environment as other experiments outlined in the paper, and both cases involve aligned small models.
>
> Within the sampling context, our method **exhibits a similar trend to the greedy decoding case**. It achieves a ~1.5x speedup without compromising performance and a 1.8x speedup with a modest 1-point BLEU/ROUGE score reduction. For a better comparison, we include the performance/latency results of speculative decoding under the same sampling scheme for comparative analysis. We will add this point in the revised version.
>
> **IWSLT**
>
> | Method | BLEU  | Norm. Latency |
> | -- | -- | -- |
> | Baseline | 39.24 | 1x  |
> | BiLD | 39.72 (+0.48) | 1.51x |
> |  | 39.26 (+0.02) | 1.63x  |
> |  | 38.27 (-0.97) | 1.80x |
> | Speculative Decoding | 39.48 (+0.24) | 1.30x |
>
> **XSum**
>
> | Method | ROUGE-L | Norm. Latency |
> | -- | -- | -- |
> | Baseline | 34.00 | 1x |
> | BiLD  | 34.34 (+0.34) | 1.22x |
> |   | 34.04 (+0.04) | 1.45x  |
> |  | 33.10 (-0.90) | 1.85x  |
> | Speculative Decoding | 32.65 (-1.35) | 1.23x  |
>
>
> ### W2. Hyperparameters
>
> Although optimal values can differ among models and tasks, they generally lie within the restricted tange across various scenarios. Notably, the **hyperparameter search has a very small overhead, as it is an evaluation-time hyperparameter** rather than a training-time one. For instance, in IWSLT with <1k val examples, the full exploration across the hyperparameter space in the paper in under 5 GPU hours. This is similar to the larger dataset’s case like XSum, whose hyperparameters can be swiftly determined using a subset of the evaluation dataset (1k). Despite the small size, it still maintains a high correlation (R^2 > 0.92) in both latency and ROUGE to the full dataset counterpart.
>
>
> ### W3. Comparison to NAT
>
> We acknowledge the value of NAT techniques and the shared objective of BiLD in improving token-level parallelism. In Section 2.1., we have elaborated on the comparison between BiLD and well-known NAT models including Levenshtein Transformer as follows:
>
> BiLD is different from NAT frameworks in that it eliminates the need for (1) from-scratch training and (2) adjustments to the training and inference processes, which is the core design principle of BiLD. This contrasts with the typical requirements of most NAT frameworks. These two attributes are becoming increasingly significant as more powerful pre-trained models become open-sourced, while the costs associated with training models from scratch become restrictive across various applications. As a result, only methods that circumvent the (1) need for from-scratch training and (2) alterations to the training pipeline can fully harness the benefits of pre-trained checkpoints, which is a feature not inherently straightforward within NAT.
>
> ### W4. HumanEval
>
> We appreciate the request to see further evaluation of our work. However, we believe that the datasets already used in this work are sufficiently challenging and fair benchmarks for assessing the effectiveness and generalizability of our framework. In fact, the small model’s performance on IWSLT/XSum are 37.47/31.23, which is significantly lower than the large model’s counterpart that are 40.33/35.10. Qualitatively, Figure 9 (Appendix 6.3.2) also visually illustrates the small model's difficulties in solving machine translation problems. As a result, these datasets have been widely adopted in previous studies [1, 2, 3] for evaluating similar frameworks.
>
> We do appreciate and agree with the opinion that HumanEval would be a even more challenging benchmark for the smaller models. Should we extend BiLD to target HumanEval, the small model will need to be enlarged in model capacity to achieve reasonable predictive performance on the new task, and in turn, the large model would be made even larger to serve as a fallback option. Such an extension is essentially a direct translation of the proposed framework to a new set of small and large models. In our work, we extensively evaluated BiLD against CALM (Figure 7, Appendix 6.2.2) and speculative decoding (Section 4.2, Global rebuttal) over four benchmarks, showing that the proposed framework can achieve substantial performance-latency improvements in the general context of collaborative use of differently sized models. Therefore, our results already demonstrate the efficacy of BiLD across a diverse set of NLP tasks, and future adapations of this work could consider coordination between larger “small” models and billion-sized “large” models to target emerging LLM tasks.
>
> ### W5. Comparison to Speculative Decoding
>
> We acknowledge the comment that our paper does not include a guarantee of an unbiased estimate. In the [global rebuttal](https://openreview.net/forum?id=EfMyf9MC3t&noteId=laUzvEbREz), however, we provided an empirical observation that BiLD can outperform [1, 2] in various scenarios, along with intuitive explanations on this.
>
> [1] Accelerating Large Language Model Decoding with Speculative Sampling
>
> [2] Fast Inference from Transformers via Speculative Decoding
>
> [3] Confident Adaptive Language Modeling

---

> > ### Comment · Reviewer_NSPU · 2023-08-18
> >
> > I appreciate the authors' clarifications in the rebuttal.
> >
> > > W1. BiLD with Sampling and comparison to Speculative Decoding
> >
> > Thanks for the added clarification and results. This will surely strengthen the submission.
> >
> > > W2. Hyperparameters
> >
> > While I understand that the evaluation-time hyperparameter can make searches more efficient, it's my belief that as we aim for a general text-based agent like ChatGPT to handle a wide range of tasks, tuning hyperparameters for each task can complicate its general application.
> >
> > > W4. HumanEval
> >
> > I remain unconvinced by the author's argument regarding the challenges of machine translation and text summarization. However, I value the results presented, especially those compared to CALM and speculative decoding.
> >
> > > Overall
> >
> > All in all, some of my concerns about the paper have been addressed, leading me to raise my score to 5.

---

### Author Rebuttal · Authors · 2023-08-10

We appreciate all the reviewers for taking the time to review our work and providing us with their valuable feedback. We provided responses to the questions that each of the reviewers has commented on.
Additionally, we present a comprehensive comparison with previous works [1, 2] as a global rebuttal, which multiple reviewers had requested in common.

---

## Comparison of BiLD and Speculative Decoding

We thank the reviewers for bringing up the concurrent work [1, 2]. Given that the paper [2] is publicly known to be accepted to ICML (4/27) 2 weeks before the NeurIPS submission deadline (and their cam-ready version was only available after 5/31), we believe that the mentioned works however should be considered as concurrent and independent work as ours, and should not be the main reason for rejection:

"Note that authors are excused for not knowing about all non-refereed work (e.g. those appearing on ArXiv). Papers (whether refereed or not) appearing less than two months before the submission deadline are considered contemporaneous to NeurIPS submissions; authors are not obligated to make detailed comparisons to such papers."

Nonetheless, **we appreciate the contributions and find it useful to discuss as a concurrent work**. Following are the reviews provided to the comments provided by the reviewers.

### 1. **Differences in methodology**

While the idea of using two models with different sizes can be deemed similar to the speculative decoding framework presented in [1, 2], we have clear distinctions in detailed methodologies.


- (1) **Distinct Prediction Rejection Approach:** The primary difference lies in how we **decide the "rejection" of predictions from the small model**. In our rollback policy, we propose to make the rejection decision based on the ‘distance’ between the small and large model predictions, which differs from the rejection sampling policy outlined in [1, 2]. While [1, 2] proposes an unbiased estimator on the large model’s prediction, Figure 2 demonstrates that combining predictions from both models through our distance-based rejection approach can surpass the exclusive utilization of the large model’s prediction probability. BiLD seeks to find and utilize this optimal performance point without introducing much runtime cost. We have a further discussion below about how our rejection policy benefits text-generation performance.
- (2) **Dynamic Window Size:** Additionally, we introduce the dynamic window size in our fallback policy. In [1, 2], the window size remains a fixed hyperparameter; however, it is also highlighted in [2] that the window size can have a noticeable impact on end-to-end latency. Our approach offers an efficient and robust solution: adjusting the window size at runtime based on the small model's confidence level in run-time. Our ablation study (Section 4.3, Figure 5) demonstrates that omitting the fallback policy and periodically transitioning control to the large model,  as proposed in [1, 2], can result in notable latency degradation.
- (3) **Model Alignment Enhancement:** Beyond the core framework, we introduce a model alignment method to align the small model’s predictions with those of the large model. This enhances the framework by reducing unnecessary rejections and can be incorporated with minimal adjustments to the training pipeline.

### 2. **Quantitative Comparison**

Furthermore, we present in [W2 of Reviewer mW2r](https://openreview.net/forum?id=EfMyf9MC3t&noteId=sb2OQpyUG0) a comprehensive quantitative comparison between our method and speculative decoding [1, 2] across two different datasets: IWSLT for machine translation and XSum for summarization. In order to ensure a fair comparison that isolates the impact of the frameworks themselves, we employ the baseline (non-aligned) small model for both BiLD and speculative decoding. We maintained the same evaluation setup and hyperparameter space that are outlined in Appendix 6.1.

The table below shows two BiLD configurations across two different tasks and two different sampling methods of greedy sampling and nucleus sampling with p=0.8, and temperature of T=1.0 as in Chen et. al. [2]. In all cases, the first row of BiLD matches latency, and the second row matches BLEU/ROUGE-L scores as speculative decoding. Across all experiments, BiLD consistently outperforms speculative decoding. It achieves either (1) notably improved BLEU/ROUGE scores of up to an additional 2 points with equivalent latency gains, or (2) superior latency gains of up to +0.4x while retaining the same score. We will include this comparison in the revised paper.
Please refer to [W2 of Reviewer mW2r](https://openreview.net/forum?id=EfMyf9MC3t&noteId=sb2OQpyUG0) for more details.



### 3. **Insights on Better Latency and Performance**

Quantitative analysis reveals a consistent trend where our method, when compared to speculative decoding [1, 2], effectively enhances both text generation quality and latency. In [W1 of wHu9](https://openreview.net/forum?id=EfMyf9MC3t&noteId=NJBmWF2n8y), we provide insights and explanations into why our approach surpasses speculative decoding frameworks.
This can be summarized as (1) **better text generation quality** due to the ensembling effect and BiLD's rejection policy that leads to higher predictive performance as well as (2) **lower end-to-end latency** due to less fallback and rollback/rejections.
Please refer to [W1 of wHu9](https://openreview.net/forum?id=EfMyf9MC3t&noteId=NJBmWF2n8y) for more details.

[1] Accelerating Large Language Model Decoding with Speculative Sampling

[2] Fast Inference from Transformers via Speculative Decoding

---

### Decision · Program_Chairs · 2023-09-21

**Decision:**

Accept (poster)

**Comment:**

This paper presents a method for swapping between a large Transformer and a small Transformer to minimize decoding costs while maintaining performance. The algorithm involves a fallback policy (which essentially invokes the large model when the small model is not confident) and a rollback policy (which overwrites the small model's predictions when they disagree with the large model's predictions). Experiments were included on a variety of generation tasks, and the method was shown to decrease decoding costs while beating various baselines. Reviewers generally agreed that the paper should be accepted, with the exception of one reviewer who felt it should be compared more directly to CALM. The authors have sufficiently addressed this concern in the rebuttal and the paper should therefore be accepted.